# Community-level respiration of prokaryotic microbes may rise with global warming

Thomas P. Smith [1]*, Thomas J.H. Thomas[1], Bernardo García-Carreras[1], Sofía Sal[1], Gabriel Yvon-Durocher [2],
Thomas Bell [1] & Samrāt Pawar [1]*

Understanding how the metabolic rates of prokaryotes respond to temperature is funda-
mental to our understanding of how ecosystem functioning will be altered by climate change,
as these micro-organisms are major contributors to global carbon efflux. Ecological metabolic
theory suggests that species living at higher temperatures evolve higher growth rates than
those in cooler niches due to thermodynamic constraints. Here, using a global prokaryotic
dataset, we find that maximal growth rate at thermal optimum increases with temperature for
mesophiles (temperature optima $\lesssim 45\,°C$), but not thermophiles ($\gtrsim 45\,°C$). Furthermore,
short-term (within-day) thermal responses of prokaryotic metabolic rates are typically more
sensitive to warming than those of eukaryotes. Because climatic warming will mostly impact
ecosystems in the mesophilic temperature range, we conclude that as microbial communities
adapt to higher temperatures, their metabolic rates and therefore, biomass-specific $CO_2$
production, will inevitably rise. Using a mathematical model, we illustrate the potential global
impacts of these findings.

[1] Department of Life Sciences, Imperial College London, Silwood Park, Ascot, Berkshire SL5 7PY, UK. [2] Environment and Sustainability Institute, University of
Exeter, Penryn, Cornwall TR10 9EZ, UK. *email: thomas.smith1@imperial.ac.uk; s.pawar@imperial.ac.uk

A general understanding of how individual organisms respond to changing environmental temperature is necessary for predicting how populations, communities and ecosystems will respond to a changing climate[1–4]. Because fundamental physiological rates of ectotherms are directly affected by environmental temperature[3,5,6], climatic warming may be expected to lead to ectotherm communities with higher metabolic rates on average[3,7]. How environmental temperature drives metabolic rates of prokaryotes (bacteria and archaea) is of particular importance because they are globally ubiquitous, estimated to comprise up to half of the planet's global biomass[8], and are the end users of the majority of net primary production[9,10]. Therefore, climate-driven changes in prokaryotic metabolic rates are expected to significantly alter ecosystem productivity, nutrient cycling and carbon flux[9–14]. Indeed, increased carbon efflux ($CO_2$ emission) has been observed in experimental measures of soil $CO_2$ loss to warming[15,16], as well as the responses of other microbial metabolic processes to increased temperature such as methanogenesis[17]. However, whether the short-term (timescales of minutes to days) thermal responses of prokaryotes can be compensated by acclimation (physiological phenotypic plasticity) or longer-term (timescales of months, years or longer) evolutionary adaptation[18–20] is currently unclear. The most recent study to investigate this idea concluded that both short- and long-term responses of ecosystem-level heterotrophic respiration were similar[21]. However, this study quantified short-term responses by aggregating day-level carbon fluxes across sites, and did not have data on the direct respiratory contributions of prokaryotes per se.

The short term, or instantaneous response of metabolic traits of individual organisms to changing temperature (the intra-specific thermal response) is typically unimodal, with the thermal performance curve (TPC) of the trait increasing with temperature up to a peak value ($T_{pk}$), before decreasing as high temperature becomes detrimental to metabolic or cellular processes[2,22] (Fig. 1). The $T_{pk}$ for maximal population growth rate ($r_{max}$, a direct measure of fitness, often used as a proxy for metabolic rate), sometimes termed the thermal optimum, is expected to correspond to the typical thermal environment in which the organism's population has evolved (the long-term response)[22,23]. The Hotter-is-Better (HiB) hypothesis posits that trait performance at $T_{pk}$ (henceforth denoted by $P_{pk}$) is also expected to increase inevitably in a similar manner to the short-term intra-specific response, because of the positive temperature-dependence of rate-limiting enzymes operating at their thermal optimum (a thermodynamic constraint), i.e., $P_{pk}$ increases with $T_{pk}$ (Fig. 1a)[22–24]. Thus this hypothesis essentially links the short-term TPC of trait performance to the longer-term performance mediated by evolution. The HiB hypothesis is also implicit in the universal temperature-dependence concept of the Metabolic Theory of Ecology (MTE)[5,6,25]. However, whether the HiB hypothesis holds across organisms and environments is a question that is still debated[24,26–28]. Deviations from a HiB pattern would indicate that either thermodynamic constraints do not exist, or are compensated for by other mechanisms. In particular, an alternative hypothesis is that natural selection acts to override thermodynamic constraints, allowing peak trait performance and fitness to be, on average, equalised across different adaptation temperatures (Fig. 1b)[24]. Intermediate scenarios are also possible, where adaptation of optimal trait performance or fitness is only partially constrained thermodynamically (Fig. 1c). Moreover, trade-offs between protein rigidity and activity at high temperatures may in fact cause hot-adapted organisms to display depressed maximal fitness ($P_{pk}$ decreasing with $T_{pk}$)[29]. Indeed, the existence of thermal constraints leading to an upper limit of prokaryotic growth rates has been shown recently[30,31]. However, a comparison of short- and long-term (HiB) responses of prokaryotic populations has neither been made nor the potential effects of responses at different timescales on ecosystem fluxes studied.

Under MTE, the global thermodynamic constraint is expected to centre n ~0.65 eV for heterotrophs[5,6]. This is the long-term, interspecific (across-species) thermal sensitivity around which species are expected to evolve, which we term $E_L$ (see Fig. 1a). Mean intraspecific thermal sensitivities (i.e., acute organism or species-level responses, here termed $E_S$) have been found to be very similar to this value, although the distribution is right-skewed with a median value of ~0.55 eV[2]. However, these values are derived from data sets which have largely or entirely excluded prokaryotes. Previous work on sub-groups of prokaryotes—cyanobacteria[32] and methanogenic archaea[17]—has indicated comparatively high thermal sensitivities which deviate from the eukaryote-derived MTE expectations. Whether this deviation from MTE is a property of prokaryotes in general has never been thoroughly tested. Given the ubiquity of prokaryotes, this is a matter of particular importance for theories applying MTE to whole ecosystems[33].

Here, we build and analyse a global data set of TPCs in bacteria and archaea to quantify general patterns in both the short-term

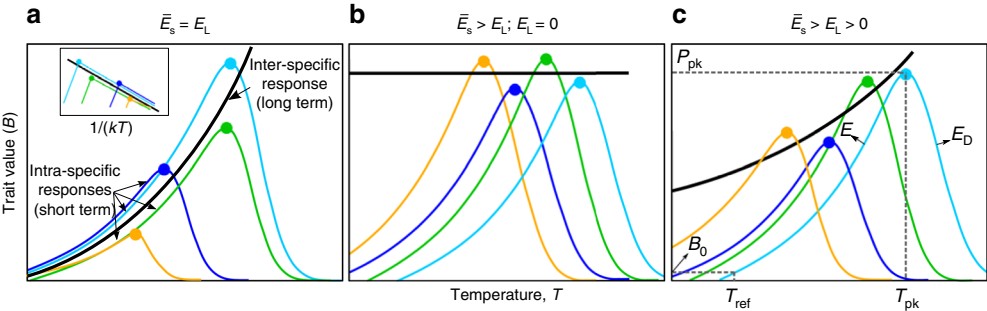

**Fig. 1** Three alternative hypotheses for short- vs. long-term thermal responses. **a** Hotter-is-Better: organisms adapt around a global, inter-specific, thermal constraint (black line, Boltzmann–Arrhenius fitted to intra-specific curve peaks), such that the average intra-specific (short-term) activation energy ($\bar{E}_S$) is statistically indistinguishable from the inter-specific (long-term) activation energy of the group of organisms ($E_L$), and both are greater than zero. See the Methods section for more details on the the definition and estimation of $\bar{E}_S$ and $E_L$, and the statistical methods used to differentiate between them. Note that each intra-specific TPC represents the short-term thermal response of each organism's population. Inset panel illustrates how this would look in an Arrhenius plot. **b** Equalisation of fitness: selection overrides thermodynamic constraints, such that trait performance at $T_{pk}$ is on average the same ($E_L = 0$). Alternatively, the same effect of $E_L = 0$ may occur due to or thermodynamic constraints on enzymes in fact restricting metabolic rate (and therefore fitness) at higher temperatures. **c** Weak biochemical adaptation: an intermediate scenario where $E_L > 0$, but significantly less than $\bar{E}_S$. Panel **c** also illustrates the the Sharpe–Schoolfield TPC model parameters (Eq. (1), Methods)

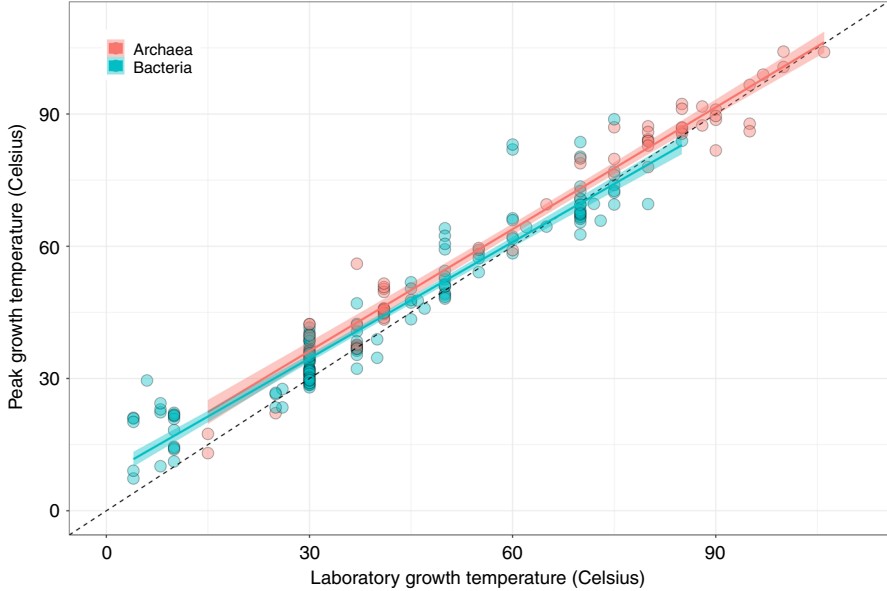

**Fig. 2** Adaptation of prokaryotic strains to lab temperatures. Peak growth temperature ($T_{pk}$) plotted against the laboratory growth temperature ($T_{lab}$), with linear models fit for bacteria (blue) and archaea (red) and a 1:1 line shown (black dashes). In general, there is a strong association of $T_{pk}$ with $T_{lab}$ (bacteria intercept $= 8.20$, slope $= 0.88$ with CI $\pm 0.04$, $R^2 = 0.92$, $p < 0.00001$; archaea intercept $= 8.59$, slope $= 0.92$ with CI $\pm 0.05$, $R^2 = 0.96$, $p < 0.00001$, linear regressions). The confidence intervals for the slopes do not include 1, however, indicating that prokaryotes tend to be unable to adapt to very high culturing temperatures. Source data are provided as a source data file

(intra-specific) response, and to test whether the HiB hypothesis holds (long-term, inter-specific response) within and across taxonomic and functional groups adapted to different temperatures (Fig. 1). These data go far beyond the scope of previous tests of the HiB hypothesis with or without microbes[24], covering practically the entire range of habitable global temperature niches (from bacteria isolated from Antarctic saline lakes at temperatures below 0 °C, to a strain of methanogenic archaea able to proliferate at 122 °C under high pressure) and the majority of the phylogenetic diversity of prokaryotes (spanning nine bacterial phyla and the two major archaeal phyla, Euryarchaeota and Crenarchaeota). In total, we compare 542 growth rate TPCs and an additional 54 metabolic flux TPCs, spanning 482 unique prokaryotic strains. In order to test the generalisations around global thermodynamic constraints, we also assemble a data set of 381 eukaryote TPCs spanning aquatic and terrestrial autotrophs, for direct comparison with our prokaryote TPCs. We find that HiB holds for mesophilic bacteria, but not thermophiles, and that mesophilic archaea also show long-term increases in rate with temperature, although the coupling of short- and long-term responses is less clear for these prokaryotes. We also show that prokaryotes tend to have generally higher thermal sensitivities than eukaryotes and conclude that as global temperatures increase, carbon efflux from the prokaryotic components of ecosystems is likely to increase at a greater rate than carbon efflux from eukaryotes.

## Results

**Adaptation to culture conditions**. First, we compared each strain's thermal optimum ($T_{pk}$) with the temperature at which it was cultured ($T_{lab}$) to determine whether the TPCs reflect adaptation to growth temperature. For both bacteria and archaea, we find a strong and significant ($p < 0.00001$, linear regression) association between $T_{pk}$ and $T_{lab}$ (Fig. 2; bacteria $R^2 = 0.92$, archaea $R^2 = 0.96$), indicating that these strains are generally well-adapted to their culturing temperature. In both archaea and bacteria data subsets, the $T_{pk}$ vs $T_{lab}$ line deviate significantly from a slope of 1 (bacteria slope $= 0.88$, 95% CI $\pm 0.04$, $n = 165$;

archaea slope $= 0.92$, 95% CI $\pm 0.05$, $n = 58$) because $T_{pk}$ tends to fall below culturing temperature at high temperatures (Fig. 2).

**Comparison of short- and long-term thermal responses**. Next, we tested the HiB hypothesis by comparing the short-term (intra-specific) and long-term (inter-specific) thermal responses (see Fig. 1; Methods). If there is a universal thermodynamic constraint, peak fitness ($P_{pk}$; $r_{max}$ at $T_{pk}$) across strains should increase with each strain's respective $T_{pk}$ (parameter $E_L$; Fig. 1) at the same rate as $r_{max}$ would increase with temperature (parameter $E_S$), on average, within single a strain's TPC. Our analysis relies on $P_{pk}$-$T_{pk}$ pairs across strains because data within strains are largely lacking, and the HiB pattern is expected to apply across strains within monophyletic taxonomic groups (such as archaea and bacteria)[24,34]. Analysing this relationship across 416 bacterial and 82 archaeal strains, we find that hotter is indeed better (HiB holds) across mesophilic bacteria ($\overline{E}_S$ and $E_L$ are $>0$, and there is significant overlap of their 95% CIs; Fig. 3 and Table 1). However, this result does not extend to thermophiles, where instead fitness is on average invariant with respect to temperature. The outcome is less clear for mesophilic archaea as whilst $E_L > 0$, there is $<50\%$ overlap of $E_L$ and $\overline{E}_S$ CIs.

**Variation in thermal sensitivity**. We find mean thermal sensitivity ($\overline{E}_S$) for bacteria $= 0.88$ eV and $\overline{E}_S$ for archaea $= 0.95$ eV. These are significantly greater than the 0.65 eV global inter-specific constraint expected under MTE[5,6] and previously observed mean intra-specific values (calculated primarily from eukaryote data)[2]. The data are right-skewed (as observed by Dell et al.[2]), but even after accounting for this skew by taking the median, activation energy still falls significantly $>0.65$ eV (bacteria median $= 0.84$ eV, archaea median $= 0.80$ eV; Supplementary Fig. 1). Furthermore, we see a consistent pattern of median thermal sensitivity $>0.65$ eV throughout the lower taxonomic groupings (Fig. 4a).

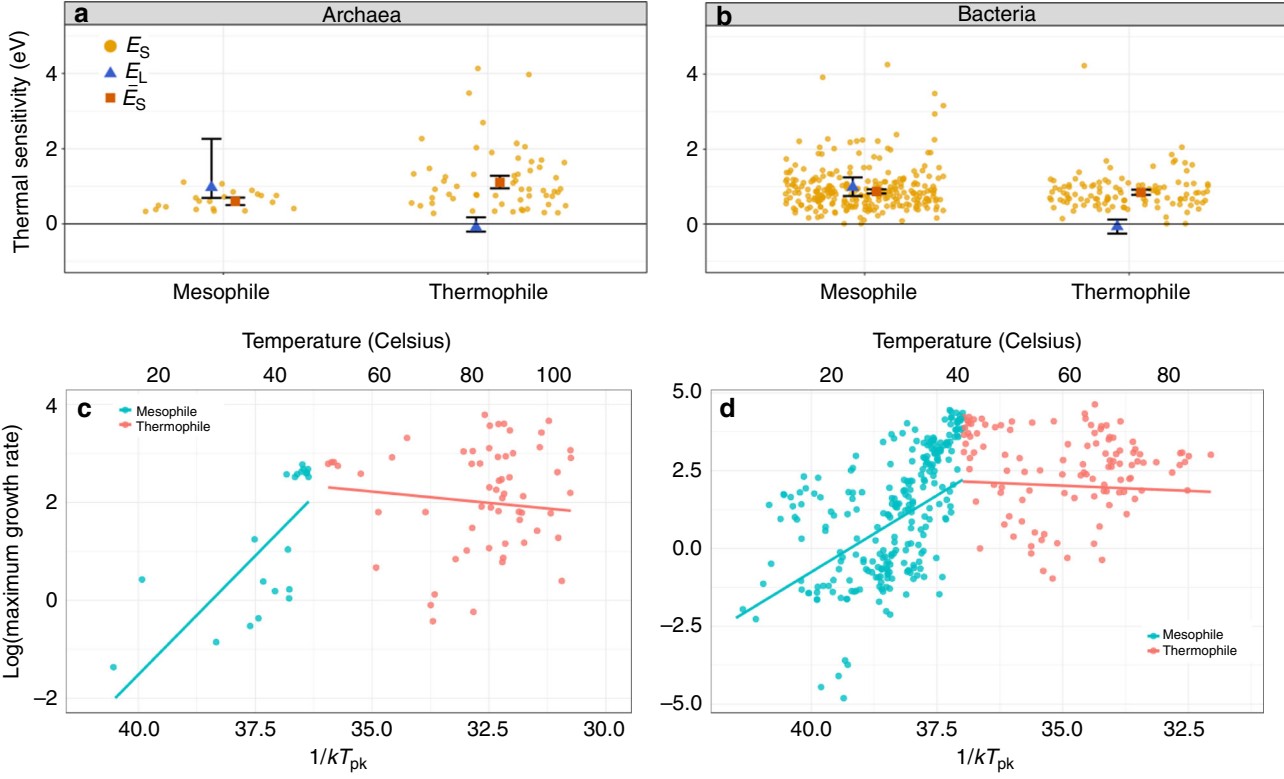

**Fig. 3** Patterns of short- and long-term thermal responses of growth rate (fitness). **a**, **b** Activation energies (with error bars showing 95% confidence intervals) from Boltzmann–Arrhenius model fits ($E_L$, blue triangles) compared with mean activation energy ($\bar{E}_S$, red squares) of the intra-specific (short-term) thermal responses. The distribution of $E_S$ is also shown (orange points). **c**, **d** Arrhenius plots ($x$-axes inverted to aid visualisation) fitted to mesophile (blue) and thermophile (red) sub-groups separately within bacteria and archaea, respectively. That is, the points are the natural log of $P_{pk}$ ($r_{max}$, per hour) plotted against $1/kT_{pk}$ (where $k$ is the Boltzmann constant) for each TPC, whilst the lines (the long-term thermal responses) are the Boltzmann–Arrhenius model fitted using weighted regression to mesophile and thermophile data separately, after determining the break point (Methods). The HiB hypothesis is best supported for the mesophilic bacteria, whilst equalisation of fitness is best supported in both thermophile sub-groups. The result for mesophilic archaea is less clear (Table 1). Source data are provided as a source data file

**Table 1 Mean short-term and long-term activation energies and test of the HiB hypothesis**

| Kingdom | Thermal niche | $n$ | $E_L$ | $\bar{E}_S$ | $E_L > 0$ | $\bar{E}_S \approx E_L$ | HiB |
|---|---|---|---|---|---|---|---|
| Bacteria | Mesophile | 264 | 0.98 (0.75–1.25) | 0.87 (0.82–0.93) | TRUE | TRUE | TRUE |
| Bacteria | Thermophile | 114 | −0.07 (−0.26–0.12) | 0.85 (0.78–0.92) | FALSE | FALSE | FALSE |
| Archaea | Mesophile | 21 | 0.97 (0.69–2.26) | 0.60 (0.50–0.70) | TRUE | FALSE | FALSE? |
| Archaea | Thermophile | 60 | −0.09 (−0.21–0.17) | 1.11 (0.95–1.28) | FALSE | FALSE | FALSE |

Estimated mean intra-specific (short-term, $\bar{E}_S$) and inter-specific (long-term, $E_L$) thermal sensitivities (95% CI ranges in parentheses) for bacteria and archaea split by thermal niche (also see Fig. 3). As the same data are used to calculate $\bar{E}_S$ and $E_L$, the number of data points ($n$) applies to both. The last column indicates whether or not the HiB hypothesis is supported. The HiB result for archaea may be ambiguous, as described in the Results and Discussion section

To further understand these findings, we investigated differences between groups of strains sharing functional traits, such as their pathogenicity and their main energy-generating metabolic pathway (Fig. 4b). Again we find mean and median thermal sensitivity >0.65 eV in the majority of functional groups, suggesting that this high $E$ is a trait generally conserved across prokaryotic organisms.

Here, we have focused on the TPCs (and activation energies) of population growth rate. However, to understand the implications of the short- and long-term thermal responses of prokaryotes for ecosystem functioning, it is necessary to test whether these reflect the activation energies of underlying metabolic flux rates. To investigate this, we assembled another thermal response data set (Methods) for metabolic fluxes recorded in prokaryotes (mainly anaerobic respiratory fluxes e.g., sulfur oxidation) and asked whether, on average, thermal sensitivity is equivalent for growth rate and metabolic fluxes. We find that average intra-specific $E$ values for growth rate TPCs were similar to, and statistically indistinguishable from the mean activation energy for metabolic fluxes (bacteria flux $\bar{E}_S = 0.82$ eV; archaea flux $\bar{E}_S = 1.01$ eV; Fig. 5a, see Supplementary Table 1 for a list of fluxes analysed). Furthermore, we compared both the prokaryotic growth rate and flux $E_S$ distributions, with a new data set (Methods) on thermal sensitivity of respiration in autotrophic eukaryotes. The results (Fig. 5d) further support a lower thermal sensitivity of short-term responses for eukaryotes than prokaryotes ($\bar{E}_S = 0.67$ eV with CI = 0.63–0.72, median = 0.57).

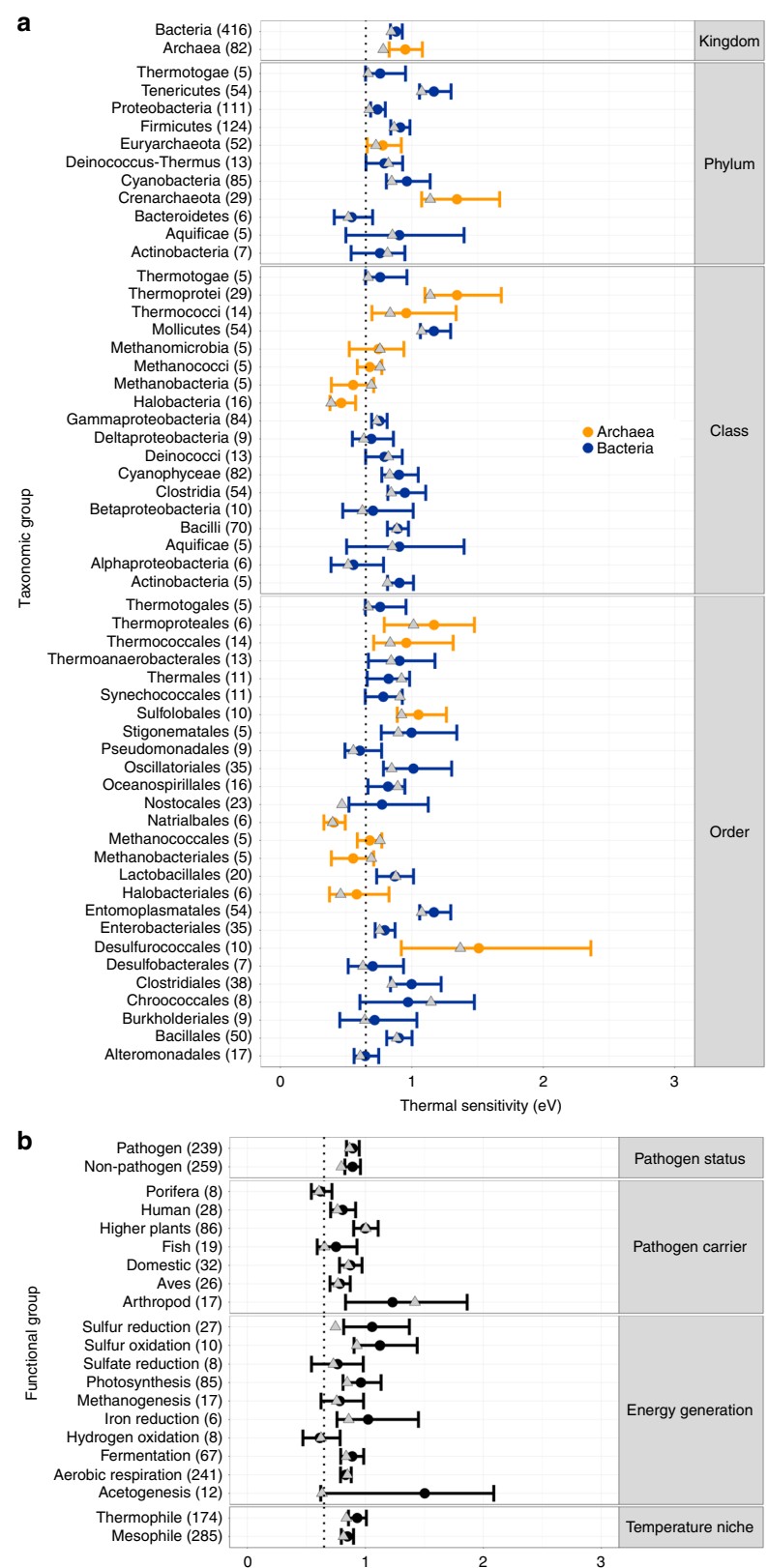

**Fig. 4** Variation in thermal sensitivity amongst prokaryotic groups. Comparison of intra-specific population growth rate activation energies ($\bar{E}_S$) across taxonomic levels (**a**) and functional trait groupings (**b**) for archaea (orange) and bacteria (blue). Points and error bars represent weighted mean and 95% CIs of $E_S$ for each group. Groups shown are those with at least five data points, the number in brackets indicates the number of data points from which $\bar{E}_S$ was calculated for each grouping. The dotted line marks 0.65 eV, the mean $E$ previously reported within the MTE framework. Grey triangles mark the median $E_S$ for each group. Source data are provided as a source data file

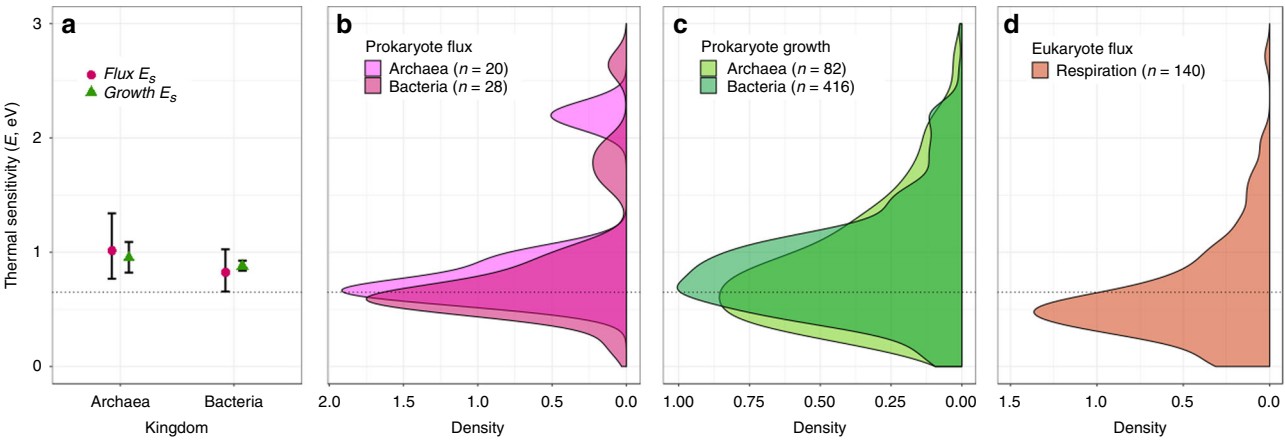

**Fig. 5** Differences in short-term thermal sensitivity across taxonomic groups. **a** Comparison of the intra-specific thermal sensitivity ($\bar{E}_S$) for growth (green triangles) and metabolic fluxes (purple circles). Error bars represent 95% CIs. CIs for growth rate thermal sensitivity fall within those for metabolic fluxes, and each sit above 0.65 eV (dotted line) for both archaea and bacteria (bacteria growth $\bar{E}_S = 0.88$ eV, $n = 416$; bacteria flux $\bar{E}_S = 0.82$ eV, $n = 28$; archaea growth $\bar{E}_S = 0.95$ eV, $n = 82$; archaea flux $\bar{E}_S = 1.01$ eV, $n = 20$). **b** Density plot of flux $E_S$ values for archaea and bacteria. **c** Density plot of growth rate $E_S$ values for archaea and bacteria. **d** Density plot of $E_S$ values for respiration rate TPCs in autotrophic eukaryotes, showing comparatively lower mean thermal sensitivity than those of the distributions for prokaryotes ($\bar{E}_S = 0.67$ eV, $n = 381$). Source data are provided as a source data file

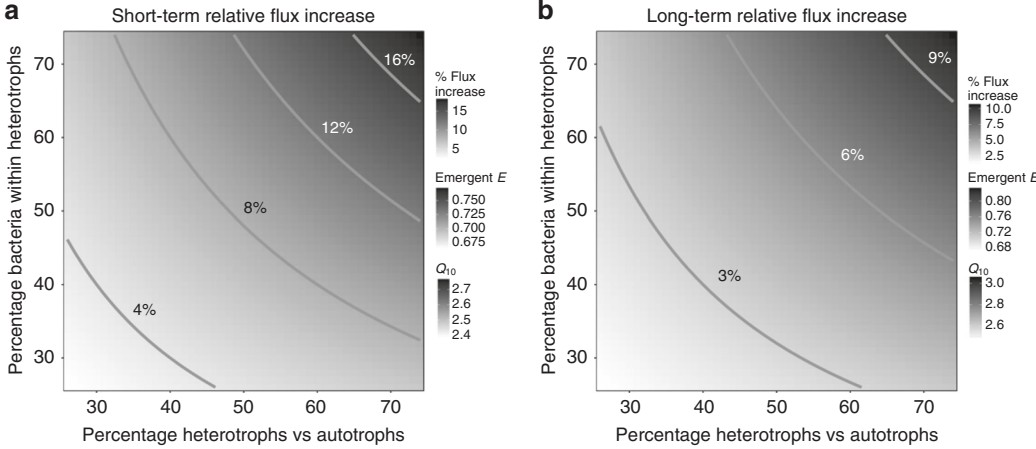

**Fig. 6** Potential changes in ecosystem carbon flux due to differences in $E$ between taxa. **a** Heatmap of % short-term increase in flux with 10 °C temperature increase (as may occur during daily fluctuations) of model ecosystems with bacteria having a different activation energy on average than eukaryotes, relative to ecosystems with all components having the same (0.65 eV) average activation energy. The flux change is shown over a range of ecosystem biomass compositions in terms of heterotrophs vs. autotrophs and bacterial proportion of the heterotrophs. The scale of emergent activation energies and $Q_{10}$s for the ecosystems with amplified flux are also shown. **b** Similar to **a**, but for long-term flux increase under a 4 °C warming scenario due to climate change. Values for the short- and long-term thermal sensitivity of bacterial thermal responses used in these calculations are our estimated $\bar{E}_S$ and $E_L$, respectively, for mesophilic bacteria (Table 1). The mathematical model is described in the Methods section

**Potential ecosystem-level impacts**. Our results suggest that higher sensitivity of both short- (higher intra-specific activation energies) and long-term (higher inter-specific activation energies —a HiB constraint) thermal responses in mesophilic prokaryotes may have profound implications for responses of ecosystem fluxes to climatic warming. To illustrate this, we built a simple mathematical model to calculate the potential change in the relative contribution of heterotrophs to ecosystem carbon efflux (based on biomass distributions typical of terrestrial forest ecosystems, see the Methods section). Using our new estimates of $\bar{E}_S$ and $E_L$ to parameterise this model, we calculate the impact of short- and long-term warming on the thermal response of carbon flux of model ecosystems that differ in composition of autotroph vs. heterotroph and eukaryote vs. prokaryote biomass. The model, consistent with previous approaches applying MTE to ecosystems, assumes that biomass remains stable with respect to changes in temperature over the timescale of the calculation (i.e.,

only mass-specific metabolic rates change in response to temperature)[33,35]. The results (Fig. 6) show that the difference in prokaryote vs. eukaryotic thermal sensitivities can substantially change the predicted increase in carbon efflux due to warming on the short- as well as long-term. For example, compared with the case where both prokaryotes and eukaryotes have the same short-term thermal sensitivity of 0.65 eV (the assumption made by most current ecosystem carbon flux models[36–38]), using the actual difference in sensitivity that we have found ($\bar{E}_S$ 0.65 eV for eukaryotes vs. 0.87 for mesophilic bacteria; Table 1; Fig. 3), the flux increases by ~8% with 10 °C short-term warming for a ecosystem composition of 50% heterotrophs (50% of which in turn are bacteria). This calculation based on the average intra-specific activation energy is relevant to short-term increases in ecosystem fluxes without evolution or acclimation in response to, for example, temperature fluctuations from timescales of minutes to days (10 °C is at the upper end of daily temperature

fluctuations that organisms may typically experience[39]). When we consider the effects of longer-term warming (such as through gradual global climate change) on the prokaryotic sub-community using the inter-specific (evolutionary) thermal sensitivity, $E_L$ (0.98 eV), we find that modelled ecosystem flux increases by ∼5% with 4 °C warming (again with 50% heterotrophs of which 50% are bacteria) compared with a baseline where the long-term thermal sensitivity is 0.65 eV for all components of the ecosystem. The actual increase in flux may indeed be higher, but is dependent upon the ratio of prokaryotic biomass to eukaryotic biomass within the ecosystem, a quantity for which estimates vary widely[8,13,40–42]. In our model, each percentage point increase in prokaryotic biomass within the heterotrophic component causes a flux increase of 0.05−0.15%, depending on the quantity of prokaryotic biomass already in the system.

## Discussion

Our results demonstrate that mean thermal sensitivities for both bacteria and archaea fall significantly >0.65 eV (Fig. 5, bacteria $\bar{E}_S = 0.88$ eV; archaea $\bar{E}_S = 0.95$ eV), suggesting that prokaryotes operate under different thermal constraints to more complex eukaryotes. Indeed, we also present a new data set of autotrophic eukaryotes which show thermal sensitivity consistent with the 0.65 eV MTE generalisation. These findings of high (relative to eukaryotes) intra-specific thermal sensitivities in prokaryotes are consistent with previous work on methanogenic archaea[17] and cyanobacteria[32], but have never been demonstrated across all major lineages of prokaryotes. In particular, Yvon-Durocher et al.[17] have argued that the high $E_S$ of methanogens are expected to translate into an increased ecosystem-level methane production at longer temporal and spatial scales. Our results suggest how these two different scales of response may be related—the short-term responses may be coupled with a HiB constraint which results in the flux at thermal optimum also increasing with (longer-term) adaptation. Moreover, this coupling across timescales is expected not just in methanogens but across most major mesophilic prokaryotes, including those involved in aerobic respiration.

The data do not allow us to determine the timescale of the adaptation resulting in the HiB pattern, but numerous previous studies have shown rapid adaptation of prokaryotes to experimental warming conditions[43–45]. Due to this adaptive capacity, as global temperatures rise prokaryotes would be expected to respond to new environmental temperatures rapidly, in effect pushing them further along the global (inter-specific) HiB curve (Fig. 1a). Alternatively, species sorting may occur such that prokaryotes inherently better-adapted to higher temperatures take advantage of temperature increases. This would have the same overall effect, because these prokaryotes would also effectively be further up the inter-specific temperature response curve (Fig. 3). In either case, under HiB, we can expect global warming to result in prokaryotic communities with higher metabolic rates on average. Whilst temperature is an important constraint, metabolic rates are also mediated by resource availability[46]. This can been seen in studies which show that carbon availability and use efficiency, community composition, and changes in microbial abundance all play roles in soil carbon loss under warming[47]. Furthermore, moisture is expected to play a significant role in microbial $CO_2$ efflux from soils[48], a factor which is itself likely to change with global warming. Thus overall, our results suggest that further production of greenhouse gases from the prokaryotic component of ecosystems is likely to increase at a greater rate than that by component eukaryotic organisms (Fig. 6), albeit mediated by other biotic and abiotic factors. Our data comprising TPCs of exponential growth rates under weak nutrient limitation may not be specific to all natural systems, however, recent work

shows that repeated assembly dynamics following perturbations are key to understanding ecosystem functioning[49]. Therefore, our empirical results and our model may be interpreted as being especially relevant to ecosystem respiration under intermittent perturbations, as would be expected in natural, open ecosystems. This also means that future work should focus on quantifying the TPCs of prokaryotic populations under different levels of nutrient limitation.

While in general, we see a tendency towards high thermal sensitivity ($E_S$) in prokaryotes, there are taxonomic sub-groups within our data set for which this is not the case (Fig. 4). For example, $\bar{E}_S$ for mesophilic archaea as a whole does not deviate significantly from the MTE 0.65 eV average (Table 1). This is largely because this subgroup is primarily comprised strains from *Halobacteria*, which have thermal sensitivities significantly <0.65 eV (Halobacteria $\bar{E}_S = 0.46$; CI = 0.38–0.58; Fig. 4a). This is likely a result of their ecologically extreme niche imposing unusual constraints on their physiology (these archaea have been isolated only from high salinity lakes). In addition, despite mesophilic archaea displaying a clear long-term increase in rate with temperature, we found only a very small overlap in CIs between $\bar{E}_S$ and $E_L$ for these strains which may not be enough to infer equivalence[50], although mapping *p*-values onto % overlap of bootstrapped CIs is not a trivial task[51]. One possibility is that mesophilic archaea follow an adaptive long-term response which is not coupled to their short-term thermal sensitivity, however, this result may also be due to shortcomings of our data set for these prokaryotes. In general, it may be harder to make generalisations about short- and long-term thermal responses across taxa for archaea as a whole, because these prokaryotes are partly typified by their propensity to adapt to different types of extreme environments[52]. That is not to say that archaea do not contribute significantly to ecosystem functioning in benign environments, however (as demonstrated through our exhaustive data collection), the thermal performance of these organisms has been generally less well-characterised. More work is necessary in order to fully understand the coupling of short- and long-term thermal responses in archaea.

We also note that while the majority of heterotrophic bacteria in our data set respire aerobically, there are a number of anaerobic strains, the majority of which were grown under various fermentation conditions. However, when we consider these groups of bacteria separately, we see no significant difference between their mean intra-specific thermal sensitivities (aerobic $\bar{E}_S = 0.86$, CI = 0.81–0.91, n = 221; fermentation $\bar{E}_S = 0.86$, CI = 0.77–0.96, n = 62). Ultimately, despite all this variation, we find that both, the short-term (intra-specific) and long-term (HiB hypothesis) amplification of metabolic rate holds true for the mesophiles (≲45 °C), temperatures which encompass most of the biomass on the planet.

Following previous approaches applying MTE to ecosystem functioning, such as Enquist et al.[35] and Schramski et al.[33], our model assumes that biomass remains constant with temperature, and therefore does not vary at the timescale of the calculation. That is, it assumes that only biomass-specific $CO_2$ efflux changes with temperature. However, realised changes to net ecosystem flux will also depend on changes in the biomass of different ecosystem components with temperature. How warming is likely to alter the overall abundances of ecosystem constituents, from microbes to plants and animals, is currently not well-understood. Future work to establish the effects of warming on population dynamics is therefore needed in order to fully understand the implications of our findings. Also, for simplicity, when parameterising our ecosystem model we used $\bar{E}_S$ and $E_G$ calculated from all of the mesophilic bacteria in the data set, as we expect a huge amount of variation in the taxa present at the ecosystem scale. Future work

can build on this by considering more specific situations where certain prokaryotes dominate in certain environments based on global biogeographic studies[53]. However, the majority of microbial taxa are known only from sequencing data[54], for example *Acidobacteria* are thought to make up in the region of 50% of soil biodiversity[55], yet very few strains have actually been cultured and therefore have TPCs available[56]. Thus in practice, it may not be feasible to accurately parameterise this sort of model based on patterns of microbial biogeography and therefore, using a global average is appropriate. Climate warming may also be accompanied by more extreme fluctuations which may be large enough to push organisms beyond their operational temperature range (OTR), within which the Boltzmann-Arrhenius model is appropriate[57]. If these fluctuations were frequent enough to have strong effects on the community, using the Sharpe–Schoolfield model as an alternative to Boltzmann–Arrhenius may be an interesting approach to predicting the effects of extreme warming events at the ecosystem level. However, parameterising this would again require the data on the specific species present in a given ecosystem and their individual TPCs, which is outside of the scope of our current study.

We have focused on the ecosystem consequences in the face of global change, but our results also have implications for understanding prokaryotic physiology. We are not aware of any previous work showing that prokaryotes differ systematically in their thermal sensitivity from eukaryotes. Therefore, further studies are needed to explore the mechanistic basis of this difference, and may reveal a major physiological transition mediated by an increase in cellular complexity as well as multi-cellularity in eukaryotes[58,59]. Also, our comparisons for growth rate and metabolic flux $E$ are simply averages across strains. Direct within-strain comparisons of growth rate (a slower thermal response) and the more instantaneous metabolic flux TPCs will be needed in order to fully understand the coupling of positive intra-specific and inter-specific thermal responses we have found here.

Our results are also important for understanding differences in thermal physiology between taxa, given our findings of HiB for mesophilic bacteria, but invariant inter-specific fitness with temperature for thermophiles (Fig. 3). Thermophiles have evolved specific adaptations to extreme temperature stress, such as mechanisms to cope with increased membrane permeability at high temperatures[60] and thus adaptation to such niches may incur a fitness cost to thermophiles as seen in our results. This result is in concurrence with an investigation of the maximum growth rates of life on Earth, which found increases in microbial growth up to a peak before an attenuation of growth rates in warmer adapted organisms[30,31]. Our results also suggest a limit to thermal adaptation as we find that strains cultured at very high temperature tend to display lower than expected thermal optima (Fig. 2). These results have implications for theoretical models of the thermal limits for life.

In summary, our results significantly deviate from current assumptions about the thermal sensitivity of heterotrophic respiration in ecosystems, and should be considered in ongoing efforts to model the impacts of climate change on ecosystem fluxes. More work needs to be undertaken to address whether intra- (short-term) and inter (long-term)-specific thermal responses are similarly conserved across other groups of organisms that are important for ecosystem function, such as fungi and insects in terrestrial, and phytoplankton and zooplankton in marine ecosystems.

## Methods

**Data collection**. We compiled a data set of published prokaryotic thermal performance curves (TPCs) by searching the literature for papers with these data and using digitisation software to collect the thermal performance point estimates.

Candidate TPC data were identified via manual searches of google scholar and pubmed databases. Search terms, such as 'bacteria', 'bacterium', 'archaea', 'archaeon', 'temperature', 'temperature response', 'thermal response', 'growth', 'adaptation', were used to find papers with response data particularly for growth rates. Later searches included terms such as 'characterisation', 'isolation', 'nov.', 'novel', 'gen.', 'sp.', as it became clear that thermal responses were often tested in publications describing newly isolated species/strains. When presented as a response curve figure, 'Plot Digitizer' software[61] was used to extract data points, including error bounds when reported. The 'Taxize' R package[62] was used to standardise taxonomy of extracted data to the NCBI database. The papers were also manually searched to collect data on growth conditions as well as other metadata where possible (historical lab growth conditions, sampling location). In instances where doubling rates or doubling times were reported, we used Doubling time $t_d = \ln(2)/\mu$ to calculate the maximum specific growth rate. Raw data were normalised to rates per second and degrees Celsius for use in modelling comparisons. In total, we collected 542 prokaryotic growth rate TPCs.

Although we primarily collected growth rate data as a measure of fitness in order to test HiB, we additionally collected 54 TPCs covering various metabolic fluxes for comparison to growth rate TPCs. Our complete prokaryote data set comprises 596 TPCs from 482 unique prokaryote strains across 239 published studies.

Finally, we compiled thermal response data for respiration rates in autotrophs from the literature using the same methods for digitisation and data collation as for the prokaryote data set. In total, this autotroph data set comprises 381 respiration rate TPCs from 140 unique autotroph species (98 vascular plants, 4 mosses, 11 green algae, 22 red algae and 5 brown algae species).

**Biological replicates and pseudoreplicates**. We use prokaryotic 'strains' to designate separate prokaryotic taxonomic entities with potentially differing TPCs. If a single study provided multiple TPCs from the same prokaryotic strain under the same conditions, these were considered pseudoreplicates. In these cases, all data were collected and a single Sharpe–Schoolfield fit (see Model fitting) was computed for the combined set of points, yielding a single set of TPC parameters. Where multiple TPCs were provided for the same strain under different growth conditions, these were considered as separate biological replicates, however in practice this is only the case for two replicates in each of two different strains in our data set. Where TPCs were obtained from prokaryotes identified only to the species level (or higher), these were considered biological replicates as likely representing different strains of those species.

The eukaryotic autotroph data set did contain organisms identified as the same species grown under similar conditions. Given the slower generation times of eukaryotes and thus slower species divergence, we do not consider these as representative of different strains, as we do for prokaryotes, and thus consider them pseudoreplicates. To remove pseudoreplicates, we compared Sharpe–Schoolfield model fits (see Model fitting) from each replicate and chose the one with the best fit as most representative of thermal performance for that species, discarding data with worse fits.

**Model fitting**. To each TPC in the data set, we fitted a modified Sharpe–Schoolfield model[63], Eq. (1):

$$B(T) = B_0 \frac{e^{\frac{-E}{k}\left(\frac{1}{T} - \frac{1}{T_{ref}}\right)}}{1 + \frac{E}{E_D - E} e^{\frac{E_D}{k}\left(\frac{1}{T_{pk}} - \frac{1}{T}\right)}} \tag{1}$$

Here, $T$ is temperature in Kelvin (K), $B$ is a biological rate, $B_0$ is a temperature-independent metabolic rate constant approximated at some (low) reference temperature $T_{ref}$, $E$ is the activation energy in electron volts (eV) (a measure of thermal sensitivity), $k$ is the Boltzmann constant ($8.617 \times 10^{-5}$ eV K$^{-1}$), $T_{pk}$ is the the temperature where the rate peaks, and $E_D$ the deactivation energy, which determines the rate of decline in the biological rate beyond $T_{pk}$. We fit this model to individual TPCs and solve for $T = T_{pk}$ to calculate the population growth rate at $T_{pk}$ ($P_{pk}$) for each strain. Note that this has been reformulated from the model presented in the original paper, to include $T_{pk}$ as an explicit parameter[64].

Each strain's TPC has a potentially different $T_{pk}$ and $P_{pk}$. Compiling these values across strains yields an inter-specific thermal response curve (Fig. 1a). TPCs without a peak are thus excluded from this analysis. We fit the Boltzmann–Arrhenius equation (Eq. (2), essentially the numerator in Eq. (1)) to these peak values to calculate inter-specific activation energy. To account for uncertainty in the original Sharpe–Schoolfield model fits to the intra-specific curves, we fitted Boltzmann–Arrhenius using a weighted regression (see accounting for uncertainty).

$$B = B_0 e^{-E/kT} \tag{2}$$

All Boltzmann–Arrhenius and Sharpe–Schoolfield model fitting were performed in Python with the NumPy package, using the Levenberg–Marquardt ordinary non-linear least-squares regression method.

**Comparing short- and long-term thermal responses**. We determined whether HiB by testing whether the activation energies from intra- (short-term) and inter-specific (long-term) TPCs were (statistically) significantly different. In order to provide a comparison between intra- ($E_S$) and inter-specific ($E_L$) activation energies, we used bootstrapping to generate confidence intervals (CIs) around the mean in each case. To provide bootstrapped CIs for $E_L$ from the modified Boltzmann–Arrhenius fits, the data were re-sampled with replacement 1000 times, with the model re-fitted to this data each time and the CIs defined as the 2.5th and 97.5th percentiles of $E$ values extracted from these fits. $\overline{E}_S$ was calculated as the weighted mean $E_S$ for the group (see methods), and CIs were taken as the 2.5th and 97.5th percentiles from the resultant distribution of $E_S$ values from a bootstrap of the weighted mean.

We then determined whether the data were consistent with either of the three hypotheses (Fig. 1) by comparing the overlap of confidence intervals of the relevant $E$ estimates. First, we tested whether $\overline{E}_S$ was greater than zero (null hypothesis that the CI includes zero). Second, we tested whether $E_L$ was greater than zero (null hypothesis that the CI includes zero). Finally, if both $\overline{E}_S$ and $E_L$ were positive, we tested whether they were significantly different to each other (null hypothesis, that the CIs for $\overline{E}_S$ and $E_L$ do not overlap). Under a HiB scenario, $P_{pk}$ will increase with $T_{pk}$ across strains, and according to MTE this is best quantified by a Boltzmann–Arrhenius model. As a result, the Boltzmann–Arrhenius activation energies from the intra- and inter-specific responses should be positive and any differences between them not statistically significant, i.e., the confidence intervals of $\overline{E}_S$ and $E_L$ should overlap each other, but not zero. Alternatively, if growth rates are not constrained by thermodynamics and $P_{pk}$ does not increase with temperature, then $E_L$ will be close to zero (CI for $E_L$ includes zero), and HiB can be rejected. Finally, in scenarios where thermodynamic constraints may be partially evident but somewhat overcome by adaptation, $\overline{E}_S$ and $E_L$ will both be positive, but with $\overline{E}_S$ being significantly greater than $E_L$ (i.e., $\overline{E}_S > E_L > 0$).

*Accounting for statistical uncertainty.* Weighted means were used to account for uncertainty in Sharpe–Schoolfield point estimates when calculating $\overline{E}_S$ and when fitting Sharpe–Schoolfield point estimates when calculating $\overline{E}_S$ and when fitting Sharpe–Schoolfield fits, we extracted the $E$ and $P_{pk}$ point estimates as well as the covariance matrix. We then sampled 1000 times from a bivariate distribution accounting for the covariance, producing 1000 model parameter combinations. We used these parameters to generate 1000 different Sharpe–Schoolfield curves, providing a distribution of $E$ and $\mu_{pk}$ from which we took the standard deviations (SD$_E$ and SD$_\mu$) as a measure of uncertainty. In some cases the Sharpe–Schoolfield fit did not produce a covariance matrix and these fits were excluded from further analysis.

When combining $E$ values across strains to calculate $\overline{E}_S$, we took weighted arithmetic means of $E$ to account for uncertainty in the original fits, where Weight $= 1/(\text{SD}_E + 1)$. Similarly, when fitting Boltzmann–Arrhenius, we apply a weighting to $\mu_{pk}$ where Weight $= 1/(\text{SD}_\mu + 1)$.

Applying these weightings does not alter the main results we obtain from this study in terms of whether the HiB hypothesis is accepted or not for different groupings, however, we felt that it was important to acknowledge and account for error in the underlying Schoolfield fits so that our results were not skewed by poor parameter estimates from questionable fits, hence this step was included. Supplementary Fig. 1 illustrates the differences between $\overline{E}_S$ calculated with and without a weighting – applying a weighting pushes $\overline{E}_S$ down a little, likely due to high $E$ values obtained from fits to lower quality data. In either case, with or without a weighting, $\overline{E}_S$ falls significantly above the 0.65 eV MTE average activation energy for both Bacteria and Archaea.

**Taxonomic and physiological groupings**. Psychrophiles and mesophiles inhabit low to medium temperature ranges, while thermophiles and hyperthermophiles grow at much higher temperatures[65]. The distinction between these groups is usually defined relatively arbitrarily, with mesophiles often considered strains with thermal optima up to 45 °C, and thermophiles those with thermal optima of 55 °C and above[65]. Corkrey et al.[30] found a peak in microbial growth rates at ~42 °C (mesophile peak) followed by an attenuation of maximum growth rates until a second peak at ~67 °C (thermophile peak), suggesting a biological transition between mesophiles and thermophiles.

In order to determine whether it was appropriate to consider mesophiles and thermophiles separately, we performed a break-point analysis on our data set using the 'Segmented' R package[66]. Segmented is not compatible with non-linear least-squares (nls) fitting, so this was performed with a linearised version of Boltzmann–Arrhenius, i.e., $x \sim y$ where $x = 1/(kT_{pk})$ and $y = \log(\mu_{pk})$. As this process was merely to confirm whether it was appropriate to split the data into mesophiles and thermophiles as suggested by eye, it is not important that these linearised fits may give slightly different slope and intercepts to the weighted nls fits. Using this methodology, we determined significant break-points for bacteria and archaea within our growth rates data set at 40.48 °C and 46.21 °C, respectively. These are similar to the ~42 °C mesophile growth rate peak seen by Corkrey et al.[30] and were thus used as cut-off points for defining mesophiles and thermophiles in our analysis. For this work, we provide no lower limit for mesophiles, essentially grouping psychrophiles and mesophiles together. Were

psychrophiles to display a pattern different to mesophiles, we would expect this to have shown up in the break-point analysis, but only one break point (separating mesophiles and thermophiles) was found for both bacteria and archaea.

In addition, archaea are typified by their adaptations to energetically demanding niches, while in contrast bacteria perform better in more ambient environments[52]. A major physiological difference between these taxa lies in their fundamentally divergent membrane structures. This affects these organisms' abilities to maintain proton gradients and thus drive metabolism under different conditions[52], a difference that may be particularly important for thermal performance. As such, we separate bacteria and archaea in our analysis as disparate organisms with divergent evolutionary histories.

In order to classify prokaryotes by the energy generating metabolic processes that they use, we took note of the growth conditions used when initially digitising the TPC data. For the majority of heterotrophic bacteria and archaea, this was simply whether they were grown under aerobic or anaerobic (fermentative) conditions. However, there are also a number of strains utilising more exotic metabolic processes such, as methanogenesis, sulfur reduction, etc. In these cases, we matched taxa against those able to utilise certain metabolic reactions according to Amend and Shock[67] before manually checking the culture conditions in each study for the metabolites required for certain metabolic processes.

We also categorised taxa by their status as potential pathogens. We matched taxon names against the database of host-pathogen interactions provided in Wardeh et al.[68] to understand whether each strain was potentially pathogenic, and what taxa they were known to infect.

**Ecosystem carbon flux model**. To quantify the effect of differences in activation energy of respiration between prokaryotes and eukaryotes on carbon flux, one can calculate the fold increase in flux ($F_x$) of an ecosystem as,

$$F_x = \frac{F_{T+x}}{F_T}, \tag{3}$$

where $x$ is the temperature increase (at the end of a warming scenario), and $F_T$ and $F_{T+x}$ are the fluxes at the two temperatures. Because ecosystem carbon flux at night (i.e., without photosynthesis) is the sum of autotrophic and heterotrophic respiration rates weighted by the biomasses of these compartments, we can re-write $F_x$ as:

$$F_x = \frac{(1-\delta)c_{ae}e^{\frac{-E_{ae}}{k(T+x)}} + \delta\left(\beta c_{hp}e^{\frac{-E_{hp}}{k(T+x)}} + (1-\beta)c_{he}e^{\frac{-E_{he}}{k(T+x)}}\right)}{(1-\delta)c_{ae}e^{\frac{-E_{ae}}{kT}} + \delta\left(\beta c_{hp}e^{\frac{-E_{hp}}{kT}} + (1-\beta)c_{he}e^{\frac{-E_{he}}{kT}}\right)}. \tag{4}$$

Here, each compartment's total flux contribution (identified by a subscript: autotrophic eukaryotes = ae; heterotrophic prokaryotes = hp; heterotrophic eukaryotes = he) is modelled as a Boltzmann–Arrhenius equation, with $c$ a normalisation constant. Each compartment's contribution is weighted by the biomass proportionality constants: $\delta$ is the proportion of heterotrophic biomass in the ecosystem, while $\beta$ is the proportion of prokaryotic biomass within the heterotrophic component (so $1 - \beta$ is the proportion of non-prokaryotic heterotrophs, such as fungi or insects). We do not use the Sharpe–Schoolfield model here because it does not apply to long-term thermal responses (Fig. 1), whilst for short-term responses most warming as well as temperature fluctuations are expected to occur within an operational temperature range, which excludes temperatures greater than $T_{pk}$ (the heat-stress region)[57]. We do not include any potential contribution of autotrophic prokaryotes (such as cyanobacteria), as these are not expected to provide a significant flux contribution to a typical terrestrial ecosystem. Alternative models and parameterisations for ecosystems which include cyanobacteria (i.e., aquatic ecosystems) are described in the Supplementary Methods and further discussed in the Supplementary Discussion.

We then use Eq. (4) to calculate the percent change in ecosystem flux due to differences in activation energies of the three compartments ($E_{ae}$, $E_{hp}$ and $E_{he}$):

$$\left(\frac{F_{x,2}}{F_{x,1}} - 1\right) * 100, \tag{5}$$

where $F_{x,2}$ and $F_{x,1}$ are the warming-induced flux changes in ecosystems with and without differences in activation energies of the compartments, respectively (the value of the heat map in Fig. 6). That is, for $F_{x,1}$, all $E$ values, i.e., $E_{au}$, $E_{hp}$ and $E_{he}$ in Eq. (4) = 0.65 eV. This is the assumption made by most current ecosystem carbon flux models[36–38]. For $F_{x,2}$, the differing activation energies were parameterised using either the mean of the estimated $E$s for the short-term (intra-specific) or the long-term (inter-specific) TPCs (Table 1; Fig. 3). For this, we used estimates of $E_L$ and $\overline{E}_S$ from mesophilic bacteria (long-term evolutionary $E_L$ = 0.98 eV, short-term instantaneous $\overline{E}_S$ = 0.87 eV) only, because the archaea in our data are largely composed of strains adapted to ecologically extreme niches, which are largely irrelevant from a global warming perspective. Archaea are known to play important roles in carbon cycling across various ecosystems however, so were more data available for the thermal performance of non-extremophilic archaea our model could be parameterised with their inclusion. Alternative model formulations to include archaea are described in the Supplementary Methods.

We calculated the emergent $E$ of the $F_{x,2}$ ecosystems (flux response to warming when prokaryotic and eukaryotic thermal sensitivities differ), which is the the average of activation energies for each ecosystem compartment weighted by its biomass proportion:

$$E = (1 - \delta)E_{ae} + \delta(\beta E_{hp} + (1 - \beta)E_{he}). \tag{6}$$

We also calculated the emergent $Q_{10}$ of the $F_{x,2}$ ecosystems, as it is a widely used measure in climate change models of carbon flux[37,69]:

$$Q_{10} = (F_{x,2})^{\frac{10}{x}}. \tag{7}$$

We chose a warming magnitude $x = 10$ °C for short-term responses because this at the upper end (e.g., generally, at higher latitudes) of the range of daily (over 24 h) fluctuations that organisms experience[39]. For long-term warming scenarios, we used $x = 4$ °C, the approximate upper end of the range for the year 2100 projected by the IPCC[70].

The biomass proportions $\delta$ and $\beta$ were varied to capture the effect of different ecosystem compositions. In a typical forest ecosystem, the contribution of autotrophic to heterotrophic (mostly soil) respiration has been estimated to be approximately 50% each[40]. This heterotrophic component would be comprised largely of prokaryotes and soil fungi biomass, the ratios of which have shown to vary widely depending on soil type and the experimental methodology used[13]. Here, we vary the percentage of heterotrophs within an ecosystem ($\delta$) between 25 and 75%, and the percentage of prokaryotes within heterotrophs ($\beta$) between 25 and 75% to generate a range of potential scenarios in Fig. 6. For simplicity, and consistent with current approaches[33,35], this model does not include changes in the relative biomass of ecosystem components with warming.

**Reporting summary**. Further information on research design is available in the Nature Research Reporting Summary linked to this article.

## Data availability

The source data underlying Figs. 2, 3, 4 and 5 and Supplementary Figs. 1 and 2 are provided as a Source Data file. All other raw data is available for download from the following git repository: https://github.com/smithtp/hotterbetterprokaryotes.

## Code availability

The data git repository also includes all of the thermal model fitting python code as well as R code to reproduce the results and figures in this manuscript, excluding figure 1 which is a conceptual diagram. https://github.com/smithtp/hotterbetterprokaryotes.

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

## Acknowledgements

We would like to thank Rebecca Kordas for helpful comments on an earlier version of the paper. TPS was supported by a BBSRC DTP scholarship (BB/J014575/1). B.G.C., S.S. and S.P. were supported by a NERC grant awarded to S.P. (NE/M004740/1). T.B. was supported by an ERC starting grant (311399-Redundancy). G.Y.-D. was supported by an ERC starting grant (677278 TEMPDEP).

## Author contributions

T.P.S. and S.P. conceived the study. T.P.S., T.J.H.T., S.S. and G.Y.-D. compiled the data. T.J.H.T. wrote the thermal model fitting code. T.P.S. and T.J.H.T. wrote the analysis code, with advice and expertise from B.G.-C. T.P.S., T.J.H.T. and S.P. analysed the data. T.P.S., T.B. and S.P. wrote the first draft of the paper, and all authors contributed to the revisions.

## Competing interests

The authors declare no conflicts of interest.
