## [Peer Review File · Nature Communications]

Reviewers' Comments:

Reviewer #1:

Remarks to the Author:

Comments on "Metabolic rates of prokaryotic microbes may inevitably rise with global warming" by Smith et al.

The authors analysed published data on prokaryotic performance/growth rate responses to temperature from pure culture studies, to infer temperature sensitivity of prokaryotes and how, consequently, their metabolic rates will change in face of climatic warming. Given the current pressure understanding and predicting how organisms and ecosystems respond to climate change, with consequences for global carbon fluxes, the topic is not only timely but also of interest to a broader audience, ranging from microbiologist, microbial ecologists, soil and marine scientists to earth system scientists. The manuscript is also well-written and the data compilation is impressive.

It's interesting to see the temperature sensitivities of prokaryotes (also on a taxonomic resolution) and I'm excited about the attempt to infer consequences for carbon fluxes from these data. However, my main (and only) concern relates to the interpretation of the results relating to consequence for ecosystem carbon fluxes. I think in several instances, the authors need to be more specific and need to consider that many factors contribute to the net response of ecosystem carbon fluxes to warming.

The authors conclude that, as microbial communities adapt to higher temperatures, their metabolic rates and therefore carbon efflux will inevitably rise (L22). For example, biomass-specific respiration rates of heterotrophic prokaryotes can increase but the net CO₂ flux may not change when the abundance (biomass) of those organisms decreases, as it has been shown for soil microbial communities. It has also been shown that after years of warming soil respiration declines and returns to pre-warmed rates. Therefore, interpretation of the results, as in L175-177, should be more conservative or specific.

The thermal performance curves for prokaryotes were determined in cultures under "optimal" growth conditions or at least under nutrient saturated conditions. One assumes that under these conditions microbes can realize a maximum growth rate at a certain temperature. However, microbes rarely have optimal nutrient conditions or non-limiting conditions in natural ecosystems. In many natural ecosystems microbial growth is energy- or nutrient- limited, or water-limited. Furthermore, an indirect effect of temperature is that increased activity will lead to substrate depletion and will consequently limit the response to temperature. So, I'm curious what the authors think about the relative importance of temperature vs. nutrient-limitation for the response of metabolic rates to climatic warming?

The authors excluded archaea from the ecosystem model because the archaea included in the study are mainly extremophiles, which are irrelevant to the focus of this study. However, I would like to point out that mesophilic archaea can contribute significantly to the prokaryotic biomass in the environment. Mesophilic archaea, comprising the phylum Thaumarchaeota (known to be ammonia oxidizers), are the most abundant and widespread environmental archaea. This group of archaea can make up a substantial part of the prokaryotic biomass in many environments, for example notably in the marine water column. Methanogenic archaea are also found in high abundances in many ecosystems, such as wetlands, rice paddy soils, deep soils, marine and freshwater sediments. Those organisms are known to be autotrophs and therefore may not be included in this model. However, the point that I want to make is that archaea are not mainly extremophiles and may still play a significant role in the carbon cycle.

L185: „and" is missing between short- and long-term

L183 & L381: Rephrase „extremophilic niche“; An organism can be extremophilic but not a niche.

L308: Comma is in the wrong place.

L356: One parenthesis too much here.

L369: I understand that autotrophic organisms are not included but “not” is missing in the sentence. It should read “We do not include...”

Reviewer #2:

Remarks to the Author:

Review of “Metabolic rates of prokaryotic microbes may inevitably rise with global warming”

In this paper, the authors assemble new datasets on the thermal performance of prokaryotic microbes for population growth and metabolic processes. They assess the relationship between the peak rates and the temperature at the peak rates to estimate a long-term thermal sensitivity across strains, and they fit a Sharpe-Schoolfield (SS) model to estimate thermal sensitivities within strains. Then they compare these sensitivities to assess competing hypotheses about thermal adaptation. They conclude the Hotter is Better (HiB) hypothesis is generally supported with also some evidence for adaptation. Finally, they use their data to parameterize an ecosystem model and predict changes in fluxes with different scenarios of warming and compositions of prokaryotic versus eukaryotic organisms.

I think this paper addresses important questions and takes a really interesting approach in comparing EL and ES values given the competing hypotheses and by showing that there are ecosystem-level implications for assuming that prokaryotes have similar thermal sensitivities to eukaryotes. I do, however, have a number of concerns about the methods and interpretation.

1. One issue to consider is the use of a BA (Boltzmann Arrhenius) equation rather than the SS or other unimodal function in the flux predictions. It is argued on line 366 that the SS is not needed because LT fluxes can be modeled by BA. I don't find this convincing. The authors have shown that they can describe the relationship between peak rate and peak temperature with BA, but not that prokaryotes don't still show unimodal TPCs after adaptation. The argument that warming will only occur up to the peak does not seem likely to hold either, as clearly climate change is accompanied by increases in temperature variance including extreme events. Even more concerning is that figure 2 shows that many bacteria have peak rates at or below a normally hot summer day in many parts of the world, indicating that a large portion of the organisms in this data set would see reduced physiological rates with further warming. Given the title of the paper and the conclusions drawn by the authors, I think it advisable to assess with the current parameterization the effect of warming using SS models in the flux equation rather than BA.

2. The correlation between culturing and T_{pk} is remarkable. I would, however, caution against interpreting this as evidence of adaptation. People would be expected to quickly identify the temperatures at which their strains go best in order to get strong growth in the lab. Thus, this curve is just as likely to be that scientists are very good at finding the best temperature to grow their cultures at as it is that cultures adapt to lab temperatures.

3. The use of confidence interval overlap might be providing a misleading view on differences between ES and EL. The issue is that when CIs don't overlap we can conclude that they are different, but when they do overlap the amount of overlap needs to be evaluated before we can conclude they are the same. This was spelled out clearly in (Cumming 2009) (and elsewhere), who showed that given a normally distributed variable, overlap up to about half the CIs still signifies a significant difference. In the case of mesophile Archaea, the EL CIs are 0.69-2.26 and the ES CIs are 0.5-0.7. Given that half the CIs are much less than the overlap, which is only 0.01, the correct conclusion here is that $EL > ES$, which does not line up with the three hypothesis in Figure 1. That makes interpretation a little tricky. However, it also means that the conclusion that this group shows HiB is not supported. This is important for the current ms as written, as HiB is one of the main take-homes.

4. I found the discussion elements scattered throughout the results to be a bit distracting. It might be

nice to remove these and keep the results a bit tidier. Lines Line 87, 98-103, and 106-108.

Smaller points from the ms:

Title – is the combo of 'may' and 'inevitably' a bit of a contradiction?

Line 32 – This is confusing. Prokaryotes are the primary consumers of plants and algae? This doesn't seem right, as most prokaryotes are picking up nutrients from the environment, involved in the break down of dead things, or autotrophic themselves. Do you mean they are the primary end user of primary production?

Line 48 – Somewhere here it might be nice to indicate a switch from saying metabolic rate is the important thing to fitness is the important thing or at least population growth and other traits are proxies for metabolic rate.

Line 57 – Might include (DeLong et al. 2018) that found evidence for HIB and its opposite in colder climates, which is another possible outcome linked to the possibilities shown in figure 1 just with $EL < 0$ for organisms adapting to cold environments.

Line 58 – Unless the thermodynamic constraints aren't there.

Lines 62-65 – I'm not sure about this statement. How do you know it's impossible for evolution to allow enzymes to function at hotter temperatures than they could before? Doesn't the existence of thermophilic bacteria kind of mean that there are ways of adapting to otherwise lethal temperatures? At the least, perhaps reword this and provide some citation support.

Line 131 – 'suggest that' after 'results'?

Line 201 – Missing an 'e' in parameterise

Lines 271-274 – Is this the same as the analysis described in the preceding paragraph, or are you doing something new?

Line 275 – Ditto above. Redundant?

Line 302 – What is mupk?

Line 320 – Is it psychrophile, with an 's'?

Line 355 – This is the fold difference not percent difference.

Line 356 – Extra parenthesis?

Line 369 – You do or don't include cyanobacteria? And your results only apply to terrestrial organisms? I imagine that many lab strains of bacteria are also aquatic?

Lines 386-389 – This is a bit unclear. Used 10C for short term warming and 4C for long term warming? This seems backwards. That 10C reflects daily variation doesn't resolve it (line 145), since it is still a figure projecting increases due to global warming, which is not a daily process. It also seems redundant – isn't Figure 5B the same as 5A just with less warming?

References

Cumming, G. 2009. Inference by eye: Reading the overlap of independent confidence intervals. *Statistics in Medicine* 28:205–220.

DeLong, J. P., G. Bachman, J. P. Gibert, T. M. Luhring, K. L. Montooth, A. Neyer, and B. Reed. 2018. Habitat, latitude and body mass influence the temperature dependence of metabolic rate. *Biology Letters* 14:20180442.

Reviewer #3:

Remarks to the Author:

This manuscript describes the impact of warming on bacteria and Archaeal growth and metabolic rates and compares them with those of autotroph eukaryotes. The authors collected information about the thermal performance curves for growth and other metabolic traits and use a model to examine the outcome of differential responses to warming on carbon efflux. While a significant amount of work is presented, some specifics are missing that makes the study a bit unclear in its present form.

I suggest several key areas of improvement before I would recommend this manuscript for

publication.

1. The abstract and introduction emphasises prokaryotes, however eukaryotes are included as comparative organisms and seem to be tacked onto the methods (i.e. we also compiled 'yet another dataset', written in Line 126-7...). Given that one of the main conclusions is that mesophilic bacteria will respond differently to warming than eukaryotes (line 175), this perceived discrepancy (about the lesser importance of eukaryotes as a focus of the study) needs to be reconciled in the way the data is currently discussed and presented.

2. This follows on from point 1. in that the authors make global comparisons of organisms yet we have no knowledge of the real ecosystems that are being explored in the model estimating carbon efflux. Line 137 states that 'model ecosystems that differ in composition of autotroph vs. heterotroph and eukaryote vs. prokaryote biomass' are being considered. It would help to have specific examples about the potential ecosystems being tested in each scenario, because this gives readers a spatial extent of the biome, and hence the significance of the findings. For example, the upper ocean is a potential test case for autotroph vs heterotroph biomass, representing 70% of the Earth's surface area.

I would also be more confident of the model outcomes if I knew organisms with sensible pairings were being compared. The eukaryotes for which data were collected include 98 vascular plants and 4 mosses, alongside 11 green algae, 22 red algae and 5 brown algae species, a weighting towards terrestrial systems. It would make sense to have microbial eukaryotes as comparative autotrophs for oceanic bacteria, but the algae listed seem to be macrophytes. More detail about whether the algae were multicellular or unicellular, from freshwater or marine habitats would help clear up the uncertainties. Furthermore, if Archaea live in generally more extreme locations (e.g., hypersaline, not necessarily hot habitats), then a more powerful test of the metabolic outcomes would be to compare bacteria, Archaea and algae from the same type of habitat. If this was done, this needs to be made clearer; if it wasn't done, then it would help to know how the authors think it may affect the model outcomes.

3. The last point is around the manuscript structure and where readers are left to find details of what was done. The main text seems to omit some important information that in some cases is found in the methods or in the Supplementary. For example, the terms are not all explained in the main text (e.g. EL), yet it's critical for conveying meaning to readers. Rather than dig around in the Methods to find essential information needed to interpret the main text, I recommend providing a brief statement in the body of the manuscript. For example, I never found a clear explanation for how short-term activation energy (ES) represents intra-specific E compared to inter-specific (long-term) activation energy (EL) – this should be cross-referenced with figure 1. The legend in figure 1 points to the methods for more information, but I found no clear definition there. Given that intra-specific typically generally means clones/strains within species, but that it may not mean the same thing in this study, it's important to clarify. Also, how many data points were collected for ES compared to EL? The place to put this detail is at line 243 under Biological replicates and pseudoreplicates and possibly in Table 1.

Specific points

Abstract

'global carbon efflux' is a rather general term; it took me a while to work out it was likely intended to be CO₂ released from the cells. While I appreciate the need to provide the 'big picture' in the abstract, a clearer statement here would provide better context, particularly for marine researchers. I noted that the use of 'carbon flux' on line 34 has a different meaning.

Results

Line 83. Regression typically tests for slopes that are different to zero, which is different to how much does a relationship deviate from 1:1. What is the test in Fig S1? To avoid confusion, it may be better to subtract Tpk from Tlab and test whether this relationship has a gradient of zero.

Line 91. I was looking for a definition of Tpk's but then realised this is intended to mean the plural of Tpk. I suggest articulating differently so that acronyms are not made plural like this.

Line 92. What is rmax? Definition is missing.

Table 1 – define the terms listed in the table.

Figure 2 – how can a maximum growth rate be negative? And what is the log (base)? Noting also that natural log of negative numbers is invalid.

Line 115. 'we also categorised the data into various groups' – is rather vague. Cross reference better to the figure and list a few examples

Line 121-2. Provide an example

Figure 4 – a bit of confusion about autotrophs – are they eukaryote autotrophs? Given photosynthetic bacteria are in the mix, I think you need to be specific.

Line 131. Word missing?

Discussion

Line 193 - why is no lower limit provided here for mesophiles?

Check spelling of parametrising; parameter

Methods

Line 233. Why rates per s?

Line 247 - cite Sharpe-Schoolfield reference

Non-linear regression of biological temperature-dependent rate models based on absolute reaction-rate theory. *Journal of theoretical biology*, 88, 719

There is no information presented on the isolation locations of the organisms in the study, so readers are left to guess...

Supplementary

Figure S2 – why are the eukaryotes not plotted here?

I hope you find my comments constructive, and that they help develop your manuscript.

Sincerely, Martina Doblin

Responses to the Reviewers

Comments made by Reviewers are in **black**, whereas our responses are in **blue**. All revisions that we have made to the manuscript mentioned or referenced below are also highlighted in the revised manuscript.

Reviewer #1

1. The authors analysed published data on prokaryotic performance/growth rate responses to temperature from pure culture studies, to infer temperature sensitivity of prokaryotes and how, consequently, their metabolic rates will change in face of climatic warming. Given the current pressure understanding and predicting how organisms and ecosystems respond to climate change, with consequences for global carbon fluxes, the topic is not only timely but also of interest to a broader audience, ranging from microbiologist, microbial ecologists, soil and marine scientists to earth system scientists. The manuscript is also well-written and the data compilation is impressive. It's interesting to see the temperature sensitivities of prokaryotes (also on a taxonomic resolution) and I'm excited about the attempt to infer consequences for carbon fluxes from these data.

We are very happy that the reviewer found our manuscript exciting and novel.

2. However, my main (and only) concern relates to the interpretation of the results relating to consequence for ecosystem carbon fluxes. I think in several instances, the authors need to be more specific and need to consider that many factors contribute to the net response of ecosystem carbon fluxes to warming. The authors conclude that, as microbial communities adapt to higher temperatures, their metabolic rates and therefore carbon efflux will inevitably rise (L22). For example, biomass-specific respiration rates of heterotrophic prokaryotes can increase but the net CO₂ flux may not change when the abundance (biomass) of those organisms decreases, as it has been shown for soil microbial communities. It has also been shown that after years of warming soil respiration declines and returns to pre-warmed rates. Therefore, interpretation of the results, as in L175-177, should be more conservative or specific.

We agree with the reviewer's assessment and have tempered the interpretation in our discussion to be more conservative (lines 193–199, also see next comment, # 3, below which follows on from this point).

3. The thermal performance curves for prokaryotes were determined in cultures under “optimal” growth conditions or at least under nutrient saturated conditions. One assumes that under these conditions microbes can realize a maximum growth rate at a certain temperature. However, microbes rarely have optimal nutrient conditions or non-limiting conditions in natural ecosystems. In many natural ecosystems microbial growth is energy- or nutrient- limited, or water-limited. Furthermore, an indirect effect of temperature is that increased activity will lead to substrate depletion and will consequently limit the response to temperature. So, I'm curious what the authors think about the relative importance of temperature vs. nutrient-limitation for the response of metabolic rates to climatic warming?

This is also a fair point, it is important to include the effects of nutrient availability as well as other biotic and abiotic factors in discussing what our results may entail for future carbon flux. We now clarify in lines 193–196 that temperature constraints are important but that other factors such as resource availability also mediate metabolic rates, and cite two new papers to support our arguments^{1,2}. We concede that optimal growth conditions

may not be applicable to all natural systems, however there are certain situations, such as repeated microbial invasions during community assembly, where maximum growth rates are likely to be important. We now add this caveat in lines 198–204 but also consider situations where our results are more relevant, citing Rivett *et al.*³

4. The authors excluded archaea from the ecosystem model because the archaea included in the study are mainly extremophiles, which are irrelevant to the focus of this study. However, I would like to point out that mesophilic archaea can contribute significantly to the prokaryotic biomass in the environment. Mesophilic archaea, comprising the phylum Thaumarchaeota (known to be ammonia oxidizers), are the most abundant and widespread environmental archaea. This group of archaea can make up a substantial part of the prokaryotic biomass in many environments, for example notably in the marine water column. Methanogenic archaea are also found in high abundances in many ecosystems, such as wetlands, rice paddy soils, deep soils, marine and freshwater sediments. Those organisms are known to be autotrophs and therefore may not be included in this model. However, the point that I want to make is that archaea are not mainly extremophiles and may still play a significant role in the carbon cycle.

We agree with the reviewer that archaea can contribute significantly to biomass and functioning in certain ecosystems. As the reviewer states, we excluded them from the model due to our data collection of archaea being mainly limited to extremophiles and thus any model parameterisation would be based on little evidence. In order to clarify this point, we have added lines 219–223 to the discussion to ensure that we do not understate the importance of archaea in ecosystem functioning. Additionally we have added supplementary work illustrating how one could add variation in archaeal abundance to the aerobic ecosystem flux model (Supplementary Methods) and we sign-post to this in the Methods (lines 443–446).

5. L185: “and” is missing between short- and long-term

Amended

6. L183 & L381: Rephrase “extremophilic niche”; An organism can be extremophilic but not a niche.

This is a fair point; the niche is extreme, and an extremophile is what lives there. We have altered the text to now refer to “ecologically extreme niches”

7. L308: Comma is in the wrong place.

Amended

8. L356: One parenthesis too much here.

Amended

9. L369: I understand that autotrophic organisms are not included but “not” is missing in the sentence. It should read “We do not include...”

Amended

Reviewer #2

1. In this paper, the authors assemble new datasets on the thermal performance of prokaryotic microbes for population growth and metabolic processes. They assess the relationship between the peak rates and the temperature at the peak rates to estimate a long-term thermal sensitivity across strains, and they fit a Sharpe-Schoolfield (SS) model to estimate

thermal sensitivities within strains. Then they compare these sensitivities to assess competing hypotheses about thermal adaptation. They conclude the Hotter is Better (HiB) hypothesis is generally supported with also some evidence for adaptation. Finally, they use their data to parameterize an ecosystem model and predict changes in fluxes with different scenarios of warming and compositions of prokaryotic versus eukaryotic organisms. I think this paper addresses important questions and takes a really interesting approach in comparing EL and ES values given the competing hypotheses and by showing that there are ecosystem-level implications for assuming that prokaryotes have similar thermal sensitivities to eukaryotes.

We are very happy that the reviewer found our manuscript interesting and novel.

2. I do, however, have a number of concerns about the methods and interpretation.

One issue to consider is the use of a BA (Boltzmann Arrhenius) equation rather than the SS or other unimodal function in the flux predictions. It is argued on line 366 that the SS is not needed because LT fluxes can be modelled by BA. I don't find this convincing. The authors have shown that they can describe the relationship between peak rate and peak temperature with BA, but not that prokaryotes don't still show unimodal TPCs after adaptation. The argument that warming will only occur up to the peak does not seem likely to hold either, as clearly climate change is accompanied by increases in temperature variance including extreme events. Even more concerning is that figure 2 shows that many bacteria have peak rates at or below a normally hot summer day in many parts of the world, indicating that a large portion of the organisms in this data set would see reduced physiological rates with further warming. Given the title of the paper and the conclusions drawn by the authors, I think it advisable to assess with the current parameterisation the effect of warming using SS models in the flux equation rather than BA.

Prokaryotes have been shown to rapidly adapt to new temperature regimes with increased fitness at previously detrimental temperatures^{4,5,6}. We agree that after adaptation their TPCs would remain unimodal, however this adaptation would also entail a shift in T_{pk} (and thus a shift in P_{pk} in line with the inter-specific hotter-is-better constraint). Therefore, after adaptation, the environmental temperatures experienced by these prokaryotes would still be below their (new) TPC peak, and thus within the range where Boltzmann-Arrhenius is an appropriate model (the "Operational Temperature Range", or OTR⁷).

Furthermore, in prokaryotes species sorting plays an important role in ecosystem-level functional "adaptation". Microbes are not strongly dispersal-limited, and community assembly has been shown to be driven by species sorting along environmental gradients⁸. Due to the vastness of microbial diversity, we would argue that as global temperature increases we may expect different (but functionally similar) species with higher T_{pk} 's to take over from less well adapted species. This again justifies the use of BA in the ecosystem model as these species would be expected to have T_{pk} greater than the typical temperature experienced in their environment (again, the OTR).

Therefore, the combination of rapid adaptation and species sorting along environmental gradients means that the inter-specific thermal response of species in a given environment can be adequately modelled using the BA model.

The reviewer makes an interesting point about climate warming being accompanied by extreme events, which may push species past their thermal optima. In the case of long-term warming effects on ecosystem carbon fluxes, the SS would still not be an appropriate model for the reasons outlined above (rapid adaptation and species sorting). For short

term (e.g., daily) warming, the extreme events would have to be large enough to push organisms beyond their OTR and frequent enough for it to have strong effects on the community. Thus, if one did decide to use the SS model to predict effects of extreme warming events on the short-term ecosystem fluxes one would need data on the species present in a given ecosystem and also their TPCs. Species-specific TPCs would be important to quantify the location of the T_{pk} 's relative to the thermal regime of the ecosystem (and determine what level of extreme events would push species outside their OTR). One could then either average their SS parameters in order to parameterise an ecosystem-level SS model, or sum up each species' contribution, the latter then also requiring data on not just species diversity but also relative abundances. All of this is outside the scope of the current study.

However, we agree with the reviewer that these are important issues, and have now addressed them in the revised discussion using the arguments we have laid out in this response (lines 239–245).

3. The correlation between culturing and T_{pk} is remarkable. I would, however, caution against interpreting this as evidence of adaptation. People would be expected to quickly identify the temperatures at which their strains go best in order to get strong growth in the lab. Thus, this curve is just as likely to be that scientists are very good at finding the best temperature to grow their cultures at as it is that cultures adapt to lab temperatures.

This is a fair point and we have added this caveat to the supplementary discussion (lines 101–104).

4. The use of confidence interval overlap might be providing a misleading view on differences between ES and EL. The issue is that when CIs don't overlap we can conclude that they are different, but when they do overlap the amount of overlap needs to be evaluated before we can conclude they are the same. This was spelled out clearly in (Cumming 2009) (and elsewhere), who showed that given a normally distributed variable, overlap up to about half the CIs still signifies a significant difference. In the case of mesophile Archaea, the EL CIs are 0.69–2.26 and the ES CIs are 0.5–0.7. Given that half the CIs are much less than the overlap, which is only 0.01, the correct conclusion here is that $EL > ES$, which does not line up with the three hypothesis in Figure 1. That makes interpretation a little tricky. However, it also means that the conclusion that this group shows HiB is not supported. This is important for the current ms as written, as HiB is one of the main take-homes.

We thank the reviewer for drawing our attention to this very important issue. We agree that there is some uncertainty in the HiB inference for archaea based on the small confidence interval overlap and have acknowledged this in the results (lines 116–117) and table 1. We previously stated in the discussion that E_S for archaea is not significantly greater than 0.65, using CIs, which also contradicts our HiB inference somewhat. We have now added to this section further discussing the apparent mismatch between E_S and E_L in mesophilic archaea (lines 211–216). This does not change the main message of our paper as both bacteria and archaea show a long-term increase in rates with warming.

5. I found the discussion elements scattered throughout the results to be a bit distracting. It might be nice to remove these and keep the results a bit tidier. Lines Line 87, 98–103, and 106–108.

We agree with the reviewer's assessment. We have reworded the 'Variation in Thermal Sensitivity' section (lines 119–122) and have redistributed other elements of this section into the introduction (lines 69–73) to also reconcile Reviewer 3's point # 1. Additionally we have revised and moved elements on thermal adaptation to the discussion (lines 255–264).

6. Title is the combo of ‘may’ and ‘inevitably’ a bit of a contradiction?

We would argue that in an age of bombastic titles for work pertaining to climate change, we have tempered ours somewhat through the usage of ‘may’. We use ‘inevitably’ in the case that our results are translatable to global patterns then increases in metabolic rates will be inevitable. We use ‘may’ as our work is not definitive and requires more understanding.

7. Line 32 This is confusing. Prokaryotes are the primary consumers of plants and algae? This doesn’t seem right, as most prokaryotes are picking up nutrients from the environment, involved in the break down of dead things, or autotrophic themselves. Do you mean they are the primary end user of primary production?

The reviewer makes a good point, prokaryotes may not directly consume plants and algae, but are the primary end-users of primary production from these organisms. We have reworded to reflect this (line 32).

8. Line 48 Somewhere here it might be nice to indicate a switch from saying metabolic rate is the important thing to fitness is the important thing or at least population growth and other traits are proxies for metabolic rate.

We now note in line 48 that fitness is often used as a proxy for metabolic rate.

9. Line 57 Might include (DeLong et al. 2018) that found evidence for HIB and its opposite in colder climates, which is another possible outcome linked to the possibilities shown in figure 1 just with $EL < 0$ for organisms adapting to cold environments.

We agree with the reviewer and have added this citation.

10. Line 58 Unless the thermodynamic constraints aren’t there.

True, added this point (lines 58–59).

11. Lines 62-65 I’m not sure about this statement. How do you know it’s impossible for evolution to allow enzymes to function at hotter temperatures than they could before? Doesn’t the existence of thermophilic bacteria kind of mean that there are ways of adapting to otherwise lethal temperatures? At the least, perhaps reword this and provide some citation support.

We appreciate the reviewer’s comment on this line and agree that it lacks some clarity. Whilst the existence of thermophilic bacteria does indeed indicate that adaptation to high temperatures is possible, the point we intended to make was that this adaptation was likely to come with a fitness cost. We have now reworded this to focus on the trade-off between protein stability and activity – enzymes need flexibility to function and maximise activity, however to function at high temperature proteins must evolve to be more rigid to prevent denaturation, citing Pucci & Rooman⁹ (lines 63–65).

12. Line 131 “suggest that” after “results”?

Amended.

13. Line 201 Missing an ‘e’ in parameterise

Amended.

14. Lines 271-274 Is this the same as the analysis described in the preceding paragraph, or are you doing something new?

We agree – these lines are largely a repetition of methods already described in the previous paragraph. We have removed the repetitive elements and merged the extra information

into the preceding section (lines 323–324 and 325–327).

15. Line 275 Ditto above. Redundant?

As above.

16. Line 302 What is μ_{pk} ?

Changed to P_{pk} , consistent with the rest of the manuscript.

17. Line 320 Is it psychrophile, with an ‘s’?

Amended

18. Line 355 This is the fold difference not percent difference.

Wording altered

19. Line 356 Extra parenthesis?

Amended

20. Line 369 You do or don’t include cyanobacteria? And your results only apply to terrestrial organisms? I imagine that many lab strains of bacteria are also aquatic?

We didn’t include cyanobacteria in this model (wording altered to reflect this more clearly). We wanted to illustrate the potential implications of our results for climate change science through a model based on terrestrial organisms. We have now added supplementary methods (referenced in the main text methods lines 430–432) which detail alternative model parameterisations for aquatic ecosystems which include cyanobacteria.

21. Lines 386-389 This is a bit unclear. Used 10C for short term warming and 4C for long term warming? This seems backwards. That 10C reflects daily variation doesn’t resolve it (line 145), since it is still a figure projecting increases due to global warming, which is not a daily process. It also seems redundant isn’t Figure 5B the same as 5A just with less warming?

In this paper we attempt to scale up from short-term to long-term responses and suggest that in order to understand the effects of climate change, it is important to understand the short-term effects of temperature on organisms.

We did use 10°C for short-term warming as this is the sort of temperature fluctuation that may occur over the period of a day and is thus apt for projecting differences in flux represented by acute population level thermal response. Panel A of figure 5 (now Figure 6 in the revised manuscript) is not intended to project differences due to global warming but differences in carbon flux projections based on daily fluctuations. We think that we describe this well in lines 153–166 of the main text and 450–453 in the methods.

The confusion may have arisen due to our use of “climate-driven” in the figure caption which is misleading. We have altered the wording of the figure caption to address this and provide more clarity of what each panel represents.

Reviewer #3

1. This manuscript describes the impact of warming on bacteria and Archaeal growth and metabolic rates and compares them with those of autotroph eukaryotes. The authors collected information about the thermal performance curves for growth and other metabolic traits and use a model to examine the outcome of differential responses to warming on

carbon efflux. While a significant amount of work is presented, some specifics are missing that makes the study a bit unclear in its present form. I suggest several key areas of improvement before I would recommend this manuscript for publication.

1. The abstract and introduction emphasises prokaryotes, however eukaryotes are included as comparative organisms and seem to be tacked onto the methods (i.e. we also compiled ‘yet another dataset’, written in Line 126-7...). Given that one of the main conclusions is that mesophilic bacteria will respond differently to warming than eukaryotes (line 175), this perceived discrepancy (about the lesser importance of eukaryotes as a focus of the study) needs to be reconciled in the way the data is currently discussed and presented.

The main focus of the manuscript is prokaryotes. However, we agree with the reviewer that the work on eukaryotes was not presented in the correct way. We have altered the wording in line 139 to make the eukaryote data seem less “tacked-on” and brought up this comparative data in the introduction (lines 88–90). We have also added lines 69–78 to the introduction to make clearer a main point of the manuscript – that while much is known about eukaryote thermal sensitivities already, prokaryotes have been largely overlooked, especially in theories looking to generalise across species, such as the Metabolic Theory of Ecology applied to ecosystems (e.g. Schramski *et al.*¹⁰).

2. This follows on from point 1. in that the authors make global comparisons of organisms yet we have no knowledge of the real ecosystems that are being explored in the model estimating carbon efflux. Line 137 states that “model ecosystems that differ in composition of autotroph vs. heterotroph and eukaryote vs. prokaryote biomass” are being considered. It would help to have specific examples about the potential ecosystems being tested in each scenario, because this gives readers a spatial extent of the biome, and hence the significance of the findings. For example, the upper ocean is a potential test case for autotroph vs heterotroph biomass, representing 70% of the Earth’s surface area.

The model we use in the main text is based on potential biomass distributions of a terrestrial ecosystem. We stated this previously in the methods, but not in the results section. We now state this in the results section as well for clarity (lines 148–149). We have also now added further supplementary methods which provide alternative model formulations and parameterisations for aquatic systems. This section is referenced in the methods of the main text (lines 430–432).

I would also be more confident of the model outcomes if I knew organisms with sensible pairings were being compared. The eukaryotes for which data were collected include 98 vascular plants and 4 mosses, alongside 11 green algae, 22 red algae and 5 brown algae species, a weighting towards terrestrial systems. It would make sense to have microbial eukaryotes as comparative autotrophs for oceanic bacteria, but the algae listed seem to be macrophytes. More detail about whether the algae were multicellular or unicellular, from freshwater or marine habitats would help clear up the uncertainties. Furthermore, if Archaea live in generally more extreme locations (e.g., hypersaline, not necessarily hot habitats), then a more powerful test of the metabolic outcomes would be to compare bacteria, Archaea and algae from the same type of habitat. If this was done, this needs to be made clearer; if it wasn’t done, then it would help to know how the authors think it may affect the model outcomes.

One of the major points of this manuscript is that *on average* prokaryotes have greater thermal sensitivity than eukaryotes. We show that, whilst there is some variation, this high thermal sensitivity is a trait shared across many taxonomic and physiological groups of prokaryotes and thus based our model on this global average.

We have added to our work by now displaying a breakdown of thermal sensitivities for the eukaryotes (Supplementary Figure 2 and Supplementary Discussion), showing that terrestrial and aquatic eukaryotes have E very similar both to each other and to the MTE 0.65eV generalisation, further strengthening our use of this parameter in the modelling work.

Whilst it would be interesting to try to pair organisms from the same habitats in order to understand variation between ecosystems with varying taxonomic compositions, we don't think that this is within the scope of this paper.

3. The last point is around the manuscript structure and where readers are left to find details of what was done. The main text seems to omit some important information that in some cases is found in the methods or in the Supplementary. For example, the terms are not all explained in the main text (e.g. E_L), yet it's critical for conveying meaning to readers. Rather than dig around in the Methods to find essential information needed to interpret the main text, I recommend providing a brief statement in the body of the manuscript. For example, I never found a clear explanation for how short-term activation energy (E_S) represents intra-specific E compared to inter-specific (long-term) activation energy (E_L) this should be cross-referenced with figure 1. The legend in figure 1 points to the methods for more information, but I found no clear definition there. Given that intra-specific typically generally means clones/strains within species, but that it may not mean the same thing in this study, it's important to clarify.

The reviewer makes a good point that these terms need more clarification. We now more directly state the meanings of E_S and E_L in the main text (lines 69–72), cross-referencing to Fig 1 as suggested. Additionally, we have added further clarification in the methods (line 334) and the caption of Table 1.

4. Also, how many data points were collected for E_S compared to E_L ? The place to put this detail is at line 243 under Biological replicates and pseudoreplicates and possibly in Table 1.

The same data is used for E_S and E_L and therefore “n” given in table 1 applies to both. We have clarified this in the table caption.

5. Abstract: ‘global carbon efflux’ is a rather general term; it took me a while to work out it was likely intended to be CO₂ released from the cells. While I appreciate the need to provide the ‘big picture’ in the abstract, a clearer statement here would provide better context, particularly for marine researchers. I noted that the use of ‘carbon flux’ on line 34 has a different meaning.

Whilst efflux is term widely used in terrestrial (especially soil) literature (e.g. Srivastava *et al.*¹¹, Kim *et al.*¹²), we agree that clarification would be helpful here for a wider audience. In the abstract we now use “CO₂ production” instead of carbon efflux, and in line 35 we add further clarification that efflux means carbon emission.

6. Line 83. Regression typically tests for slopes that are different to zero, which is different to how much does a relationship deviate from 1:1. What is the test in Fig S1? To avoid confusion, it may be better to subtract T_{pk} from T_{lab} and test whether this relationship has a gradient of zero.

We simply ask whether the confidence intervals of the slopes of the regression lines include 1. This is fairly standard practice for asking whether regression slopes deviate from each other, see DeLong *et al.*¹³ for example. We now make clearer in the figure legend that this was the test and provide further description in the supplementary methods. A revised

version of this figure now appears as figure 2 in the main text.

7. Line 91. I was looking for a definition of Tpk_s but then realised this is intended to mean the plural of Tpk. I suggest articulating differently so that acronyms are not made plural like this.

We have reworded this (line 108).

8. Line 92. What is r_{max}? Definition is missing.

Added a definition for r_{max} earlier in the manuscript (line 48).

9. Table 1 define the terms listed in the table.

Added definitions to the caption

10. Figure 2 how can a maximum growth rate be negative? And what is the log (base)? Noting also that natural log of negative numbers is invalid.

This is the natural log of the maximum growth rate, negative values are the natural log of growth rates between 0 and 1. We have added to the caption to clarify. Note that this is now Figure 3 in the revised manuscript.

11. Line 115. ‘we also categorised the data into various groups’ is rather vague. Cross reference better to the figure and list a few examples

We have reworded this and provided examples (lines 126–127).

12. Line 121-2. Provide an example

Done

13. Figure 4 – a bit of confusion about autotrophs – are they eukaryote autotrophs? Given photosynthetic bacteria are in the mix, I think you need to be specific.

Good point, this could be confusing to the reader. We have altered the legend title in panel D to read “Eukaryote Flux” and now refer in the figure caption to “autotrophic eukaryotes”. Note that this is now Figure 5 in the revised manuscript.

14. Line 131. Word missing?

Amended.

15. Line 193 - why is no lower limit provided here for mesophiles?

For simplicity, we grouped psychrophiles and mesophiles together as these groupings are usually defined relatively arbitrarily with no biological reasons for a temperature cut-off separating them. Were psychrophiles to display a pattern different to mesophiles, we argue that this would have shown up in the break-point analysis. However only one break-point was found to separate both archaea and bacteria into two temperature groups, which we term “mesophiles” and “thermophiles”. We have reworded line 230 to no longer say ‘range’ as this may have implied upper and lower temperature bounds. We have also noted this in the methods (lines 392–395).

16. Check spelling of parametrising; parameter

Amended throughout the manuscript.

17. Line 233. Why rates per s?

We needed to standardise the data collected from different species via different methodology all into the same units in order for us to compare them. We standardised to seconds

as this is the basic SI unit for time. To display our results (i.e. fig. 3 C+D) we convert to more biologically appropriate units (growth rate per hour).

18. Line 247 - cite Sharpe-Schoolfield reference Non-linear regression of biological temperature-dependent rate models based on absolute reaction-rate theory. *Journal of theoretical biology*, 88, 719

We have added a sign-post to the subsequent model fitting methods section, where the Sharpe-Schoolfield model is described and cited (line 300).

19. There is no information presented on the isolation locations of the organisms in the study, so readers are left to guess...

We collected location data where easily available in the manuscripts supplying TPC data. This can be found in the raw data which we provide. However, the isolation locations of many strains are not readily available, or refer only to laboratory culture collections which provides another layer of complexity in finding the original source. As prokaryotes display patterns of global biogeography (see Louca *et al.*¹⁴, specifically “Most prokaryotic OTUs are globally distributed”) location data is in any case not likely to be informative. Additionally, many of our strains are from thermophiles which come from environments where latitude is unrelated to environmental temperature (e.g. hydrothermal vents, hot springs). We therefore decided not to pursue attempting to access location data for every strain in our dataset as it would entail much effort for likely little gain.

20. Figure S2 why are the eukaryotes not plotted here?

We now include the eukaryotes in this supplementary figure as suggested. Note that this now appears as supplementary figure S1.

References

1. Rhee, G.-Y. & Gotham, I. The effect of environmental factors on phytoplankton growth: Temperature and the interactions of temperature with nutrient limitation. *Limnology and Oceanography* **26**, 635–648 (1981).
2. Melillo, J. M. *et al.* Long-term pattern and magnitude of soil carbon feedback to the climate system in a warming world. *Science* **358**, 101–105 (2017).
3. Rivett, D. W. *et al.* Elevated success of multispecies bacterial invasions impacts community composition during ecological succession. *Ecology Letters* **21**, 516–524 (2018).
4. Bennett, A. F., Dao, K. M. & Lenski, R. E. Rapid evolution in response to high-temperature selection. *Nature* **346**, 79–81 (1990).
5. Kishimoto, T. *et al.* Transition from positive to neutral in mutation fixation along with continuing rising fitness in thermal adaptive evolution. *PLoS Genetics* **6**, 1–10 (2010).
6. Blaby, I. K. *et al.* Experimental evolution of a facultative thermophile from a mesophilic ancestor. *Applied and Environmental Microbiology* **78**, 144–155 (2012).
7. Pawar, S., Dell, A. I., Savage, V. M. & Knies, J. L. Real versus Artificial Variation in the Thermal Sensitivity of Biological Traits. *The American Naturalist* **187** (2016).
8. Van der Gucht, K. *et al.* The power of species sorting: Local factors drive bacterial community composition over a wide range of spatial scales. *Proceedings of the National Academy of Sciences* **104**, 20404–20409 (2007).

9. Pucci, F. & Rooman, M. Physical and molecular bases of protein thermal stability and cold adaptation. *Current Opinion in Structural Biology* **42**, 117–128 (2017).
10. Schramski, J. R., Dell, A. I., Grady, J. M., Sibly, R. M. & Brown, J. H. Metabolic theory predicts whole-ecosystem properties. *Proceedings of the National Academy of Sciences* **112**, 2617–2622 (2015).
11. Srivastava, P., Raghubanshi, A. S., Singh, R. & Tripathi, S. N. Soil carbon efflux and sequestration as a function of relative availability of inorganic N pools in dry tropical agroecosystem. *Applied Soil Ecology* **96**, 1–6 (2015).
12. Kim, Y., Park, S. J., Lee, B. Y. & Risk, D. Continuous measurement of soil carbon efflux with Forced Diffusion (FD) chambers in a tundra ecosystem of Alaska. *Science of the Total Environment* **566-567**, 175–184 (2016).
13. DeLong, J. P., Okie, J. G., Moses, M. E., Sibly, R. M. & Brown, J. H. Shifts in metabolic scaling, production, and efficiency across major evolutionary transitions of life. *Proceedings of the National Academy of Sciences of the United States of America* **107**, 12941–12945 (2010).
14. Louca, S., Mazel, F., Doebeli, M. & Parfrey, L. W. A census-based estimate of Earth's bacterial and archaeal diversity. *PLoS biology* **17** (2019).

Reviewers' Comments:

Reviewer #1:

Remarks to the Author:

The authors carefully revised the manuscript. However, I still have concerns regarding one of my comments. Maybe I was not clear before. I commented earlier that, despite higher metabolic rates under warming, the net CO₂ flux of an ecosystem may not increase or remain higher over years/decades of warming because microbial abundance/biomass can decrease in response. Indeed the authors stated that now somewhat in L193-199. However, that should be made clear in the context of the ecosystem model. I understand that the fluxes are weighted by the biomass proportion but I do not see how the biomass of the different compartments varies with warming. I also don't think that this is necessary here. But absolute changes in biomass of plants, heterotrophic prokaryotes and eukaryotes, are important when it comes to absolute/net fluxes.

I understand that the model shows the change in the relative contribution to the C efflux but not the absolute ecosystem flux, as the size of the different components do not vary. In its current form, the manuscript reads as the net CO₂ flux will increase with warming. I think only a few amendments are necessary, to make clear that the ecosystem model does not show absolute ecosystem fluxes. Just to give an example: Heading of Figure 6: "Short-term and long-term ecosystem flux increase". It could be replaced by e.g., "Increase in relative contribution to C efflux". Would that be correct? I also suggest to amend the abstract accordingly: L22 with "cell- or biomass-specific CO₂ production rates" or something similar.

If the authors carefully revise the manuscript accordingly, it will prevent readers getting the wrong take home message about the ecosystem consequence of global warming from this work.

Reviewer #4:

Remarks to the Author:

I was asked to step in here in round 2 since two reviewers from round one were unable to participate. I think the first round of reviewers did an excellent job, so rather than introduce a new set of concerns or comments here, I will focus on evaluating reviewer responses to the original set of concerns.

In particular, I focused on the balance needed in extrapolating from culture based studies to ecosystem consequences. Generally, I am comfortable with this in the revisions. One important note-warming can also lead to drier soils, in which case moisture availability and diffusion are likely to become limiting factors rather than temperature.

Second, I focused on reviewer 2's comments about the appropriate model. I thought the reviewers response was well reasoned, and the modification to the paper appropriate.

In sum, this is a nice paper, and in my view the reviewer comments have been addressed.

Responses to the Reviewers

The second round of comments made by Reviewers are in **black**, whereas our responses are in **blue**. All revisions that we have made to the manuscript mentioned or referenced below are also highlighted in the revised manuscript.

Reviewer #1

1. The authors carefully revised the manuscript. However, I still have concerns regarding one of my comments. Maybe I was not clear before. I commented earlier that, despite higher metabolic rates under warming, the net CO₂ flux of an ecosystem may not increase or remain higher over years/decades of warming because microbial abundance/biomass can decrease in response. Indeed the authors stated that now somewhat in L193-199. However, that should be made clear in the context of the ecosystem model. I understand that the fluxes are weighted by the biomass proportion but I do not see how the biomass of the different compartments varies with warming. I also don't think that this is necessary here. But absolute changes in biomass of plants, heterotrophic prokaryotes and eukaryotes, are important when it comes to absolute/net fluxes.

This is a fair point and we have now clarified for the reader, both in the results (lines 147-148 and 152-154) and methods (lines 473-475), that for simplicity (consistent with previous work applying MTE to ecosystems^{1,2}) the model does not include variation in biomass with respect to temperature at the timescale of the calculation. That is, biomass changes much more slowly than mass-specific metabolic rates in response to temperature. This point is further addressed in the next comment below.

2. I understand that the model shows the change in the relative contribution to the C efflux but not the absolute ecosystem flux, as the size of the different components do not vary. In its current form, the manuscript reads as the net CO₂ flux will increase with warming. I think only a few amendments are necessary, to make clear that the ecosystem model does not show absolute ecosystem fluxes. Just to give an example: Heading of Figure 6: "Short-term and long-term ecosystem flux increase". It could be replaced by e.g., "Increase in relative contribution to C efflux". Would that be correct? I also suggest to amend the abstract accordingly: L22 with "cell- or biomass-specific CO₂ production rates" or something similar. If the authors carefully revise the manuscript accordingly, it will prevent readers getting the wrong take home message about the ecosystem consequence of global warming from this work.

This comment is related to the above comment #1; we agree that this is an important point that didn't come across properly in the manuscript previously. For short term temperature changes (on the daily scale), we would not expect significant changes in biomass. However, long term warming is, as the reviewer states, likely to be associated with changes in biomass of different ecosystem components, as they respond and adapt to climate change. We have made the following amendments to clarify to the reader that our model does not include variation in biomass and so cannot predict net ecosystem flux, but predicts changes in the relative contribution of different components:

We have altered the abstract to include "biomass-specific" as suggested (line 23), making a minor wording change to line 20 to accommodate this within the word limit.

We clarify in the discussion (lines 236-243) that these results are specific to changes in biomass-specific CO₂ efflux and discuss further the gaps in knowledge around changes in biomass with warming.

We have altered the headings of figure 6 to read “Short-...” and “Long-term relative flux increase”

Reviewer #4

1. I was asked to step in here in round 2 since two reviewers from round one were unable to participate. I think the first round of reviewers did an excellent job, so rather than introduce a new set of concerns or comments here, I will focus on evaluating reviewer responses to the original set of concerns. In particular, I focused on the balance needed in extrapolating from culture based studies to ecosystem consequences. Generally, I am comfortable with this in the revisions. One important note- warming can also lead to drier soils, in which case moisture availability and diffusion are likely to become limiting factors rather than temperature.

The reviewer makes a good point about moisture availability which we had omitted. This caveat is now addressed in the discussion (lines 198-200), citing Hawkes *et al.*³.

2. Second, I focused on reviewer 2’s comments about the appropriate model. I thought the reviewers response was well reasoned, and the modification to the paper appropriate. In sum, this is a nice paper, and in my view the reviewer comments have been addressed.

We appreciate the time taken by the reviewer to evaluate this work.

References

1. Enquist, B. J. *et al.* Scaling metabolism from organisms to ecosystems. *Nature* **423**, 639–642 (2003).
2. Schramski, J. R., Dell, A. I., Grady, J. M., Sibly, R. M. & Brown, J. H. Metabolic theory predicts whole-ecosystem properties. *Proceedings of the National Academy of Sciences* **112**, 2617–2622 (2015).
3. Hawkes, C. V., Waring, B. G., Rocca, J. D. & Kivlin, S. N. Historical climate controls soil respiration responses to current soil moisture. *Proceedings of the National Academy of Sciences* **114**, 6322–6327 (2017).